# Breeders that receive help age more slowly in a cooperatively breeding bird

Martijn Hammers [1], Sjouke A. Kingma [1,2], Lewis G. Spurgin[3], Kat Bebbington[1,3], Hannah L. Dugdale [4], Terry Burke [5], Jan Komdeur[1] & David S. Richardson[3,6]

Helping by group members is predicted to lead to delayed senescence by affecting the trade-off between current reproduction and future survival for dominant breeders. Here we investigate this prediction in the Seychelles warbler, *Acrocephalus sechellensis*, in which mainly female subordinate helpers (both co-breeders and non-breeding helpers) often help dominants raise offspring. We find that the late-life decline in survival usually observed in this species is greatly reduced in female dominants when a helper is present. Female dominants with a female helper show reduced telomere attrition, a measure that reflects biological ageing in this and other species. Finally, the probability of having female, but not male, helpers increases with dominant female age. Our results suggest that delayed senescence is a key benefit of cooperative breeding for elderly dominants and support the idea that sociality and delayed senescence are positively self-reinforcing. Such an effect may help explain why social species often have longer lifespans.

[1] Groningen Institute for Evolutionary Life Sciences, University of Groningen, 9712 CP Groningen, The Netherlands. [2] Department of Animal Science, Wageningen University & Research, 6708 PB Wageningen, The Netherlands. [3] School of Biological Sciences, University of East Anglia, Norwich NR47TJ, UK. [4] School of Biology, University of Leeds, Leeds LS29JT, UK. [5] Department of Animal and Plant Sciences, University of Sheffield, Sheffield S102TN, UK. [6] Nature Seychelles, Victoria, Mahé, Seychelles. These authors jointly supervised this work: Jan Komdeur, David S. Richardson. Correspondence and requests for materials should be addressed to M.H. (email: m.hammers@rug.nl)

Variation in ageing patterns observed across taxa is enormous, but the causes of this variation are still poorly understood[1]. Intriguingly, even within the same species there is often extensive individual variation in the onset and rate of actuarial senescence—the progressive age-dependent decline in survival[2]. Elucidating the causes of among-individual variation in senescence is crucial to our understanding of the mechanisms and trade-offs that drive ageing within and across species. Patterns of sociality contribute significantly to explaining variation in ageing rates across species[3,4]. However, while studies have investigated relationships between intraspecific competition and senescence[5,6], studies investigating the relationship between sociality and senescence at the intraspecific level are rare and the direction of causality of this relationship remains to be resolved[7,8].

In cooperative breeding systems, parental care is generally shared between socially dominant individuals and (often related) subordinate helpers[9]. The alloparental care provided by helpers often allows the dominants to reduce their current reproductive investment, which may then reduce the negative impacts of reproductive effort on the condition of dominants (e.g. through reducing oxidative stress[10,11]) and increase the survival of helped dominants[12–16]. The benefits of having helpers are predicted to be greater for young dominants, because young dominants may have little breeding experience[17,18], and for elderly dominants, because elderly individuals may suffer greater costs of reproduction due to senescent declines in physiological condition[19]. Hence, for elderly dominants a key benefit of receiving help might be that it delays the onset, and reduces the rate, of actuarial senescence. However, studies testing whether helping alleviates actuarial senescence in dominants are lacking (but see ref. [20]). If the benefits of receiving help increase with a dominant's age, there should be a strong incentive for elderly dominants to recruit and retain helpers. Therefore, we predict that the likelihood of having helpers increases with age in dominants.

In this study, we use 15 years of data on the facultative cooperatively breeding Seychelles warbler *Acrocephalus sechellensis* to study associations between actuarial senescence and cooperative breeding. The Seychelles warbler population on Cousin Island provides a useful model system in which to study this as individuals are followed throughout their entire lives;[21] The majority of individuals (>96% since 1997) have been individually colour-ringed and are monitored annually, and interisland dispersal is extremely rare, allowing for accurate measures of survival[22–24]. Senescent declines in survival occur in the Seychelles warbler[25,26], but whether helpers offset such late-life

declines in survival has not yet been investigated. In the Seychelles warbler, telomere shortening—a measure that has been shown to reflect biological ageing in various organisms[27,28]—predicts survival[26] and reflects physiological costs[29–31]. Each year, about half of the ca 115 territories contain one to five sexually mature subordinates in addition to the dominant breeding pair. Some (20% of males and 42% of females (this study)) of these subordinates act as helpers and provide alloparental care (max. three helpers per territory) and assist in incubation (females only) and provisioning offspring[32,33]. In response to being helped, dominants reduce their incubation attendance (this study) and provisioning rate[34] (but see ref. [32]), but still gain increased reproductive success[35]. As the majority of helpers are female, only female helpers incubate, and provisioning rates of female helpers are generally higher than those of male helpers[35–37], we expect dominants to benefit more from having female helpers.

Here, we test the prediction that for dominants a reduced rate of both actuarial senescence and telomere shortening is associated with having helpers, especially female helpers. We also test whether the likelihood of having female helpers increases with age in dominants. We find that dominant females benefit from having female helpers in terms of delayed senescence and reduced telomere attrition. In addition, we find that older female, but not male, dominants are more likely to have female helpers. Our results suggest that delayed senescence is a key benefit of cooperative breeding for elderly female dominants, and support the idea that sociality and delayed senescence are positively self-reinforcing. Such an effect may help explain why social species often have longer lifespans than non-social species.

## Results

**Incubation attendance**. Female dominants with a female helper had 21% lower incubation attendance (Supplementary Fig. 1; Supplementary Table 1; mean ± SE = 39.9% ± 1.8% of time incubating, $n = 69$) than dominant females without a female helper (mean ± SE = 50.4% ± 0.8%, $n = 277$) and the total incubation attendance at a nest was 45% higher for nests with female helpers (Supplementary Table 1; mean ± SE = 73.2% ± 2.0%, $n = 69$). Incubation attendance was not associated with age of the dominant female (Supplementary Table 1).

**Helping and actuarial senescence**. Survival was strongly age-dependent and declined progressively among elderly dominants

**Table 1 Age-dependent survival of dominants in relation to helper presence**

| (a) Dominant female | | | | | (b) Dominant male | | | |
| Variable | Estimate | SE | z | P | Estimate | SE | z | P |
|---|---|---|---|---|---|---|---|---|
| Intercept | 1.98 | 0.14 | 13.84 | <0.001 | 1.73 | 0.17 | 10.20 | <0.001 |
| Age | −0.58 | 0.25 | −2.28 | 0.023 | 0.01 | 0.18 | 0.08 | 0.936 |
| Age² | −0.67 | 0.23 | −2.93 | 0.003 | −0.60 | 0.22 | −2.80 | 0.005 |
| Territory quality | 0.41 | 0.17 | 2.44 | 0.015 | 0.05 | 0.16 | 0.33 | 0.742 |
| Helper (Y/N) | −0.16 | 0.24 | −0.65 | 0.513 | 0.41 | 0.22 | 1.90 | 0.057 |
| Number of subordinates | 0.22 | 0.19 | 1.18 | 0.239 | −0.30 | 0.16 | −1.95 | 0.051 |
| Age × helper | 1.25 | 0.39 | 3.18 | 0.001 | 0.64 | 0.35 | 1.82 | 0.069 |
| Age × number of subordinates | −0.12 | 0.36 | −0.33 | 0.740 | −0.11 | 0.28 | −0.41 | 0.685 |

| Random | Variance | 1571 Observations | Variance | 1581 Observations |
|---|---|---|---|---|
| Individual ID | 0.27 | 463 Individuals | <0.01 | 491 Individuals |
| Year | 0.11 | 15 years | 0.29 | 15 years |

Final models contained all main effects and significant interaction terms

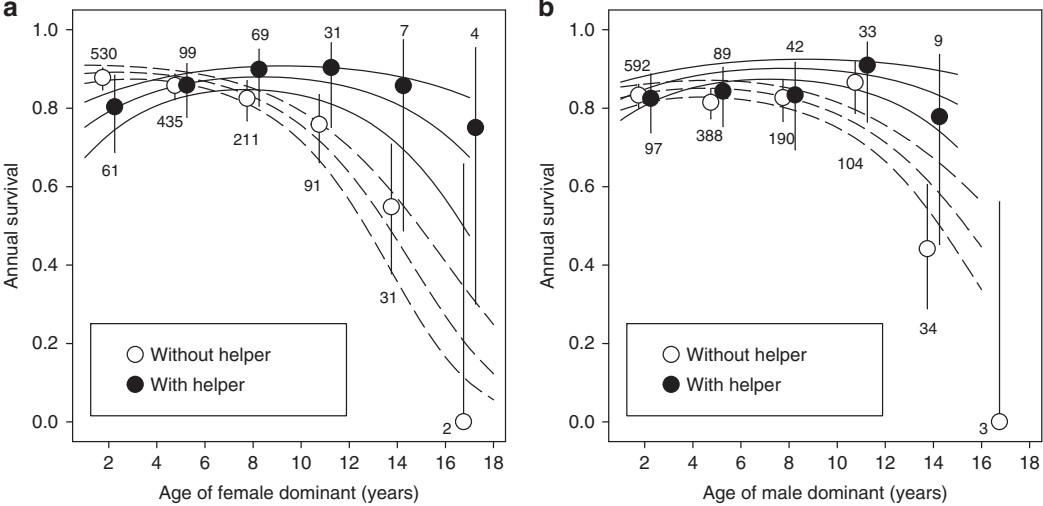

**Fig. 1** Age-dependent survival of dominants in relation to helper presence. **a** Dominant females, **b** dominant males. Solid lines are model predicted slopes ± SE for dominants that were assisted by helpers during the main breeding season and dashed lines are for dominants without helpers. Data shown are means (circles) and 95% binomial confidence intervals (error bars) for 3-year age intervals (e.g. 1–3 year) based on raw data. In the analyses, age was a continuous variable. Numbers are sample sizes. Source data are provided as a Source Data file

of both sexes (Table 1; Fig. 1). When averaged across all ages, annual survival probabilities of female dominants without helpers (84%) and with helpers (86%) were similar (two proportion $z$-test: $\chi^2 = 0.29$, $P = 0.590$). However, the impact of helpers of either sex on dominant female survival depended on the dominant female's age. Survival of female dominants that were not assisted by helpers declined strongly with age, but the survival of dominants that received help showed little age-dependence and the late-life decline was much less pronounced (Table 1; Fig. 1). Indeed, survival of female dominants < 7 years old (i.e. before the onset of reproductive senescence in this species) was similar for individuals with (84%) or without (87%) a helper of either sex (two proportion $z$-test: $\chi^2 = 0.87$, $P = 0.352$), but among elderly dominants (>6 years) survival was higher for dominants with helpers (89%) than for dominants without (78%; two proportion $z$-test: $\chi^2 = 6.40$, $P = 0.011$), which is due to a decline in survival of elderly dominants without helpers (Fig. 1). The effect of helpers on survival was independent of the number of subordinates (helpers and non-helpers) that were present in the territory, or its interaction with age (Table 1). This indicates that helping by subordinates, rather than (factors associated with) the presence of subordinates, predicted the age-related survival effect in dominants. A model that included two separate binary variables for female and male helper presence instead of presence of a helper of either sex was less well supported by the data ($\Delta AICc = 3.9$), but suggested that the age-dependent impact of helper presence on dominant female survival is mainly explained by the presence of female helpers (Supplementary Table 2). We did not find an association between dominant female (age-dependent) survival and male helper presence (Table 1), though the likelihood of detecting such an effect is reduced because male helpers are much less common than female helpers, especially among elderly dominants (Supplementary Fig. 2).

Similar to the impact of helpers on the age-dependent survival of female dominants, we found some evidence for an association between (female) helper presence and age-dependent survival of male dominants, although this was not, or only marginally, statistically significant (helper of either sex × dominant male age, GLMM: $P = 0.069$: Table 1, Fig. 1; female helper × dominant male age, GLMM: $P = 0.049$, Supplementary Table 2). Again, a

model that included two separate binary variables for female and male helper presence instead of presence of a helper of either sex was less well supported ($\Delta AICc = 2.8$). When averaged across all ages, the annual survival probabilities of male dominants without helpers (82%) and with helpers (84%) were similar (two proportion $z$-test: $\chi^2 = 0.71$, $P = 0.400$). Survival of male dominants < 7 years old was similar for individuals with (83%) or without (83%) a helper of either sex (two proportion $z$-test: $\chi^2 = 0.02$, $P = 0.879$). Among elderly dominants (>6 years) survival tended to be higher for male dominants with helpers (86%) than for male dominants without (79%), although this difference was not significant (two proportion $z$-test: $\chi^2 = 1.44$, $P = 0.230$).

**Telomere attrition rate.** The within-individual rate of attrition of telomeres (ΔRTL) differed between dominant females with and without a female helper (Table 2; Fig. 2), or a helper of either sex (Supplementary Table 3). The number of subordinates that was present in a territory also predicted ΔRTL in dominant females, but this effect was in the opposite direction to that observed for helper presence (Table 2). We then tested whether ΔRTL was below zero in unassisted dominant females and above zero in dominant females with a female helper. ΔRTL declined in unassisted dominant females (Fig. 2; one-sided $t$-test: $t_{37} = -2.27$, $P = 0.015$), but the apparent increase in ΔRTL in dominant females with a helper was not significant (Fig. 2; one-sided $t$-test: $t_6 = 1.27$, $P = 0.125$). For dominant males, ΔRTL was not associated with female helper presence (Table 2; Fig. 2).

**Age-dependent helper prevalence and subordinate reproduction.** Overall, older (≥2 years old) subordinates were more likely to help (mean ± SE = 0.56 ± 0.03, $n = 292$) than younger (≤1 year old) subordinates (mean ± SE = 0.23 ± 0.02, $n = 647$), and female subordinates (mean ± SE = 0.42 ± 0.02, $n = 571$) more than male subordinates (mean ± SE = 0.20 ± 0.02, $n = 368$; Table 3). The likelihood that a subordinate helped was associated with the age of the dominant female, but the direction of this association was dependent on the subordinate's sex: positive for female subordinates and negative for male subordinates (Table 3; Fig. 3). Indeed, among female dominants with a helper, the likelihood

**Table 2 Annual change in relative telomere length (RTL) in dominants in relation to female helper presence**

| | (a) Dominant female | | | | (b) Dominant male | | | |
|---|---|---|---|---|---|---|---|---|
| Variable | Estimate | SE | t | P | Estimate | SE | t | P |
| Intercept | −0.20 | 0.05 | −4.30 | <0.001 | −0.07 | 0.07 | −0.95 | 0.351 |
| Initial RTL | −0.72 | 0.07 | −10.23 | <0.001 | −0.56 | 0.08 | −6.74 | <0.001 |
| Age | 0.15 | 0.07 | 2.08 | 0.044 | −0.08 | 0.09 | −0.86 | 0.397 |
| Territory quality | 0.00 | 0.07 | −0.04 | 0.971 | 0.12 | 0.08 | 1.46 | 0.150 |
| Offspring produced (Y/N) | 0.03 | 0.07 | 0.38 | 0.709 | 0.08 | 0.08 | 0.96 | 0.341 |
| Female helper (Y/N) | 0.45 | 0.12 | 3.73 | <0.001 | 0.03 | 0.10 | 0.25 | 0.802 |
| Number of subordinates | −0.31 | 0.09 | −3.44 | 0.001 | −0.01 | 0.09 | −0.12 | 0.905 |

| Random | Variance | 45 Observations | Variance | 74 Observations |
|---|---|---|---|---|
| Individual ID | <0.01 | 39 Individuals | 0.02 | 58 Individuals |
| Cohort | <0.01 | 18 Cohorts | 0.02 | 16 Cohorts |
| Year | <0.01 | 11 years | <0.01 | 9 years |
| Residual | 0.05 | | 0.07 | |

Final models contained all main effects and significant interaction terms

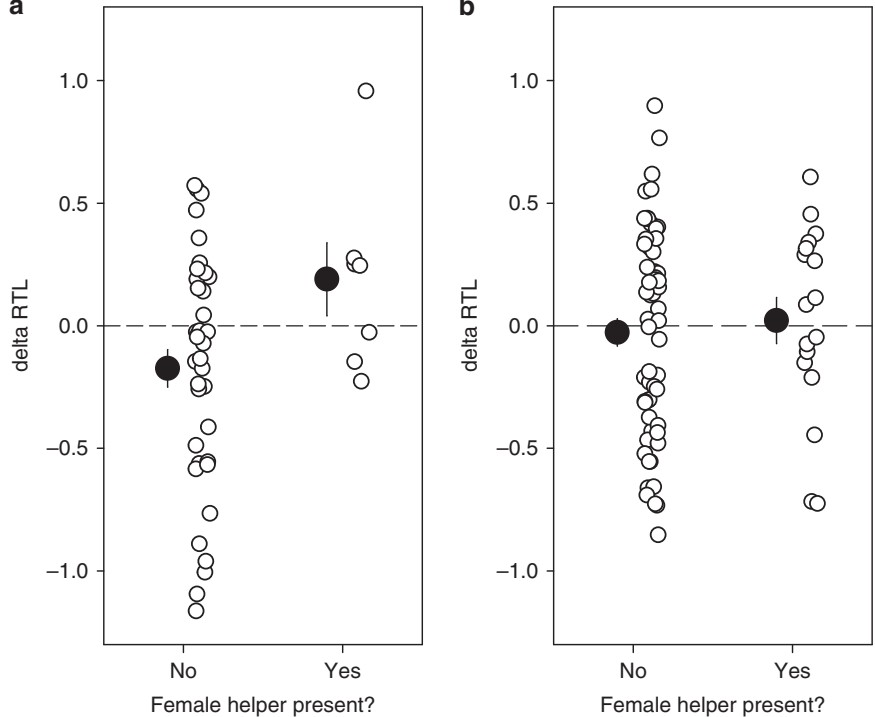

**Fig. 2** Annual change in relative telomere length (delta RTL) in dominants in relation to female helper presence. **a** Dominant females, **b** dominant males. The dashed line indicates no telomere shortening or lengthening. Filled circles are means and s.e.m. of raw data, open circles are raw data points. Source data are provided as a Source Data file

that a helper was female increased with the female dominant's age (Table 4). As a result, elderly dominants almost exclusively had female, but not male, helpers (Fig. 4). Neither the likelihood that a male or female subordinate helped, nor the sex ratio among helpers, were related to the dominant male's age (Tables 3, 4; Figs. 3, 4).

The likelihood that a subordinate female reproduced was higher for older (≥2 years old) subordinates (0.30 ± 0.03, mean ± SE, $n = 227$), while younger (≤1 year old) subordinates almost never reproduced (0.02 ± 0.01, mean ± SE, $n = 344$) (Supplementary Table 4). Subordinate reproduction was not related to the age of

the dominant female or male, territory quality and the number of subordinates that was present in the territory (Supplementary Table 4).

## Discussion

Sociality might play a key role in explaining some of the considerable inter-specific and intraspecific variation in senescence observed in nature[3,4], but it is currently unclear whether social phenomena like alloparental care can truly affect senescence patterns, or whether senescence can explain variation in social behaviour. In this study, we found that while the survival of

**Table 3 The likelihood that a subordinate helped in relation to the dominant's age and the subordinate's sex**

| (a) Dominant female | | | | | (b) Dominant male | | | |
|---|---|---|---|---|---|---|---|---|
| Variable | Estimate | SE | z | P | Estimate | SE | z | P |
| Intercept | −1.09 | 0.21 | −5.14 | <0.001 | −1.12 | 0.21 | −5.31 | <0.001 |
| Dominant age | 0.43 | 0.22 | 1.98 | 0.048 | 0.04 | 0.19 | 0.23 | 0.821 |
| Territory quality | 0.10 | 0.23 | 0.42 | 0.674 | 0.14 | 0.23 | 0.61 | 0.543 |
| Subordinate sex (male) | −1.05 | 0.21 | −5.10 | <0.001 | −0.95 | 0.20 | −4.80 | <0.001 |
| Subordinate age (older) | 1.73 | 0.24 | 7.26 | <0.001 | 1.85 | 0.24 | 7.62 | <0.001 |
| Number of subordinates | −0.27 | 0.20 | −1.38 | 0.167 | −0.29 | 0.19 | −1.50 | 0.134 |
| Dominant age * Subordinate sex | −1.00 | 0.41 | −2.43 | 0.015 | 0.07 | 0.37 | 0.18 | 0.858 |

| Random | Variance | 939 Observations | Variance | 929 Observations |
|---|---|---|---|---|
| Group ID | 0.48 | 673 Groups | 0.29 | 671 Groups |
| Individual ID | 0.16 | 294 Individuals | 0.30 | 302 Individuals |
| Year | 0.28 | 15 years | 0.27 | 15 years |

Final models contained all main effects and significant interaction terms

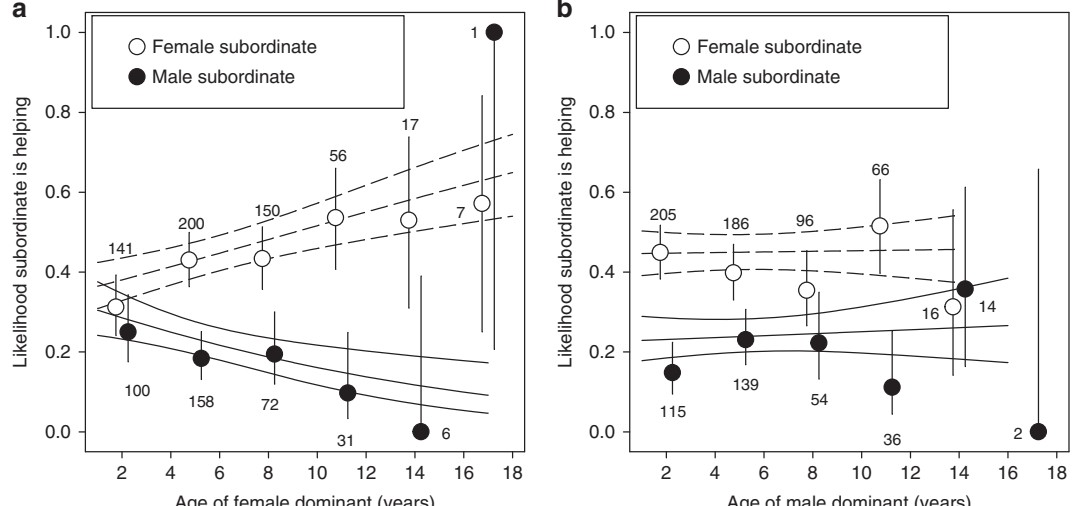

**Fig. 3** The likelihood that a subordinate helped in relation to the dominant's age and the subordinate's sex. **a** Dominant females, **b** dominant males. Solid lines are model predicted slopes ± SE for male subordinates and dashed lines are for female subordinates. Circles with error bars are means and binomial 95% confidence intervals for 3-year age intervals based on raw data for female subordinates (open circles) and male subordinates (filled circles). Numbers are sample sizes. Source data are provided as a Source Data file

unassisted elderly dominants of both sexes declined progressively with age, the age-specific decline in survival of female dominants was greatly reduced if they were assisted by helpers. We also found that helper presence was associated with reduced telomere shortening (a marker of biological ageing in this and many other species[27,28]) in dominant females, but not in dominant males. In addition, we found that elderly female, but not male, dominants were more likely to have female helpers and less likely to have male helpers. In other words, our results suggest that helpers may contribute to delay senescence in female dominants and that, at the same time, dominant females acquire more female helpers as they get older.

In cooperatively breeding species, dominants often show higher survival when assisted by helpers[14–16,38], but an absence of survival differences between individuals with and without helpers is also frequently observed in cooperatively breeding birds[16]. The finding that in the Seychelles warbler only elderly individuals, that normally have lower survival because of senescence, benefit from

receiving help, could be caused by a ceiling effect: the very high annual survival in young and mid-aged individuals means there is little potential for improvement in survival, but there is much more scope for this in elderly individuals with lower survival probabilities. Another explanation may be that the costs of reproduction, or maintaining a territory, only become apparent in individuals suffering senescence, not in younger individuals that are in better physiological condition[19].

Survival benefits for dominants can arise because helpers allow dominants to reduce their costs of reproduction, thereby allowing them to invest more resources in somatic maintenance[12,13]. For example, helpers may reduce the costs of incubation and investment in eggs for the dominant female[15,39]. In dominant female Seychelles warblers, incubation costs are probably lower for those that have a helper as assisted females reduce their incubation attendance by 21% (this study), while hatching success increases[40]. The fact that we only detected reduced telomere shortening in female dominants with female helpers, but not in

**Table 4 The likelihood that a helper is a male in relation to the age of the dominants**

| (a) Dominant female | | | | | (b) Dominant male | | | |
|---|---|---|---|---|---|---|---|---|
| Variable | Estimate | SE | z | P | Estimate | SE | z | P |
| Intercept | −0.82 | 0.25 | −3.25 | 0.001 | −0.62 | 0.22 | −2.77 | 0.006 |
| Dominant age | −1.07 | 0.42 | −2.54 | 0.011 | 0.27 | 0.33 | 0.82 | 0.411 |
| Territory quality | 0.16 | 0.36 | 0.44 | 0.662 | 0.24 | 0.35 | 0.68 | 0.494 |
| Subordinate age (older) | −1.47 | 0.36 | −4.04 | <0.001 | −1.73 | 0.36 | −4.79 | <0.001 |
| Number of subordinates | −0.07 | 0.36 | −0.20 | 0.841 | −0.08 | 0.35 | −0.23 | 0.820 |
| **Random** | **Variance** | **310 Observations** | | | **Variance** | **309 Observations** | | |
| Group ID | <0.01 | 271 Groups | | | <0.01 | 270 Groups | | |
| Individual ID | 0.86 | 156 Individuals | | | 0.64 | 162 Individuals | | |
| Year | <0.01 | 15 years | | | <0.01 | 15 years | | |

Final models contained all main effects

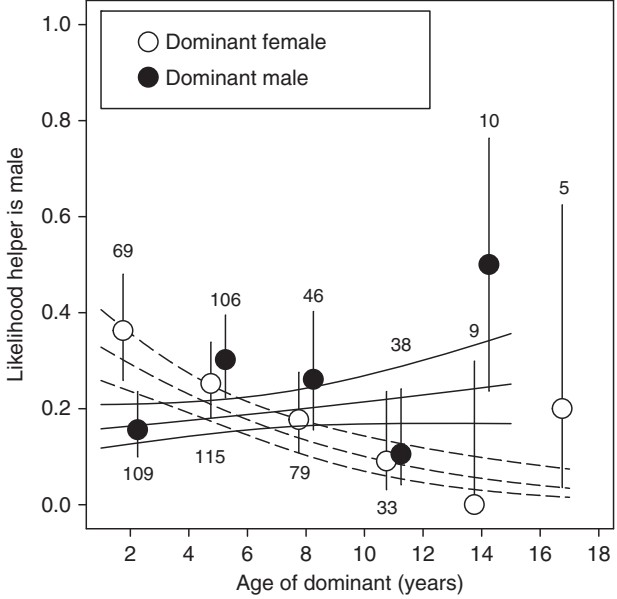

**Fig. 4** The likelihood that a helper is a male in relation to the age of the dominants. The dashed lines are the model predicted regression slope ± SE for female dominants and the solid lines are those for dominant males. Circles with error bars are means and binomial 95% confidence intervals for 3-year age intervals based on raw data. Numbers are sample sizes. Source data are provided as a Source Data file

male dominants, is perhaps explained by the fact that only females incur significant costs of incubation in this species, and these costs are alleviated by female helpers.

The telomere results contrast with our analysis of survival, where we found that the presence of female helpers appears to have similar effects on age-specific survival of both dominant females and males, although this was only significant for dominant females. This suggests that benefits other than reducing the costs of incubation play a role in survival. Intriguingly, we also observed that telomere lengths tended to increase in female dominants that were assisted by female helpers (although this increase was not significant), but declined in unassisted female dominants. Although measurement error may explain some observations of telomere length increasing over time within individuals[41], there is increasing evidence that actual telomere length increases do occur[42], in which telomerase expression may

play a key role[43]. Of particular relevance, there is clear evidence that telomere lengthening occurs in the Seychelles warbler[31], where mortality has been linked to (shorter) telomere length[26].

Our analyses suggest that the presence of helpers, rather than of subordinate group members per se (which is often challenging to separate in cooperatively breeding species[44]), explains the higher late-life survival of dominants with helpers. A limitation of our study, and of most other studies on cooperatively breeding species[44], is that we cannot easily disentangle the impact of help from the quality or condition of the dominants. For example, better quality individuals with longer lifespans and higher reproductive output might be more likely to have helpers because they have successfully reproduced in previous years. However, the impact of helpers on survival persisted when the number of subordinates was also included in the models, which suggests that our results are not simply explained by differences in individual quality or territory quality. Moreover, the greater telomere shortening (a longitudinal measure across two points within each female) observed in female dominants that were not helped, compared to the lack of shortening in helped females, suggests that helpers prevent a deterioration of the dominant female's condition, rather than that dominants with a helper were initially in better condition (of better quality). Future studies should attempt experimental manipulations of the amount of help that dominants receive to confirm the causality of the associations found in our study. Experimental manipulations will also help to test the possibility that subordinates are more likely to help when they assess the dominants as being in better physiological condition or more productive.

The survival benefits of receiving help may reduce the fitness costs of senescence in elderly individuals. Elderly dominant female Seychelles warblers show a drop in reproductive output during the last year of life and the magnitude of this drop increases with age[45]. If having helpers allows dominants to postpone their death, this may compensate for the decline in reproductive output, and enhance the dominant's late-life reproductive performance. If help is beneficial for elderly dominants, then dominants might offer subordinates incentives to stay and help (e.g. food, protection or opportunities to reproduce), and try to retain subordinates that they would normally have evicted from their territory. Subordinates may benefit from increased survival of the dominants as this enhances the indirect fitness benefits received by related subordinates[37,46] and survival of the subordinates[33]. It also provides female subordinates with an opportunity to gain direct benefits in the form of co-breeding[35]. Here we do not tease apart whether the effects outlined above arise from co-breeding or alloparental helping as

separating these two types of helpers is difficult in this system, given that some non-breeders may be individuals that have attempted to breed, but failed to do so successfully. Co-breeding also provides an additional benefit for helpers as they can share reproduction and parental care with the dominants (and then thus have 'helpers' themselves). In addition, previous studies found that having a (co-breeding) helper is beneficial for the dominant's reproductive success[35,40]. Therefore, it seems likely that both helpers and dominants benefit from each other, perhaps especially when elderly dominants are suffering senescence.

The logic outlined above leads to the intriguing possibility that elderly individuals might be able to use cooperative breeding as a strategy to increase their lifespan and to maximize lifetime reproductive success. We found some evidence that this might be the case in the Seychelles warbler. Although we do not know the actual mechanism, the likelihood that subordinates helped increased with the age of the dominant female. This increase in helper prevalence was explained by an age-dependent increase in female (not male) helper prevalence, resulting in increasingly female-biased helper sex ratios in territories with elderly dominant females (from ca 60% in younger dominants to nearly 100% in elderly dominants). We can only speculate why we only found this relationship for dominant females, but a potential explanation may be that dominant females benefit most from female helpers because they invest more in reproduction. In turn, female subordinates might have more incentive to stay and help by an offer of a share in reproduction[40]. Although the likelihood that a female subordinate reproduced appeared to be unrelated to the age of the dominant female or male, future studies should test whether co-breeding frequencies increase—and eviction rates, or levels of aggression towards subordinates, decrease—among elderly dominants in this species. A thorough examination of the direct and indirect benefits for dominant and subordinate group members is required to test whether there may be positive reinforcement between dominants living longer because of the help of subordinates, and subordinates being more likely to stay and help when receiving more benefits when assisting elderly dominants.

In the longer term, helping-enabled improvements in the late-life survival of dominants may drive the evolution of longer lifespan in cooperative breeders, but this prediction remains to be tested. Some comparative studies found no association between longevity and cooperative breeding across bird species[47,48] (but see ref. [49]). A possible explanation for this is that the impact of receiving help on senescence might differ strongly between species. This could occur if the strength and direction of this relationship depends on the species' ecology or life-history strategy. Another explanation that remains to be tested is that helping delays actuarial senescence and leads to longer lifespans in the receivers of help, but that the mean lifespan across the population remains similar because helpers show accelerated senescence and shorter lifespans. Furthermore, because the force of natural selection is proportional to the number of individuals alive in a given age class[50], the small number of elderly dominants that benefit from help (Fig. 1) means that selection on delayed senescence may be relatively weak compared to factors that improve fitness during early life. However, a positive effect of helpers on the dominant's fitness in late life should nonetheless select for delayed senescence and longer lifespan in dominants, and thus increased cooperative breeding.

Our results suggest that for elderly dominants, higher late-life survival may be a key benefit of cooperative breeding. More studies investigating how helping affects senescence at the individual level are needed to test how the association between cooperative breeding and senescence differs between the individual and species level. We encourage future studies to investigate how cooperation may delay senescence, how the prevalence of cooperation may change with age, and whether cooperation and delayed senescence may be self-reinforcing[21,51–53], thus potentially driving longer lifespans in social species.

## Methods

**The Seychelles warbler model system.** The Seychelles warbler population on the isolated island of Cousin (29 ha; 4°20' S, 55°40' E) contains ca 320 adult individuals, nearly all of which are colour-banded (using a combination of three colour rings and a British Trust for Ornithology metal ring)[54]. The warbler's life history is characterized by high annual adult survival (84%), mostly single-egg clutches, and extended periods (up to three months) of post-fledging care[24,32]. Individuals that have acquired a dominant breeding position generally defend the same territory, with the same partner, until their death[55]. The correlation between the age of the dominant male and female in a territory is, while significant, actually relatively weak (Pearson product-moment correlation: $r = 0.16$, $t_{1531} = 6.53$, $P < 0.001$, Supplementary Fig. 3). This is because the age at which an individual obtains a dominant position varies considerably, pairs of birds do not become dominant at the same age, and the age at which dominant individuals die (and one of the pair is replaced) varies. Previous studies have shown that male and female dominants have similar breeding tenure, annual survival probabilities and rates of actuarial senescence[24,25]. The vast majority of breeding activity occurs in June–September (hereafter: main breeding season), when food availability is highest (breeding occurs in 94% of territories in this period)[56]. Seychelles warblers can breed successfully in socially monogamous pairs, but, because of a lack of suitable breeding opportunities, young individuals often delay independent breeding and become subordinates within a territory, where they then may help with providing alloparental care (incubation (female subordinates only); provisioning (male and female subordinates)), or not[54]. Subordinates are often retained offspring from previous breeding attempts[33], although a very small number of subordinates disperse to a new subordinate position in a different territory[57]. Territory inheritance in the Seychelles warbler is rare (only 3.7% of dominant breeding positions are obtained via offspring inheriting this status on their natal territory[58]), so it is unlikely that inheritance is the main benefit accrued by subordinates. Subordinates benefit from helping as they obtain breeding experience[59] and often gain indirect (kin-selected) fitness benefits through helping related offspring[46]. Further, older (≥2 year old) female subordinates often (ca 40% in any year) gain direct fitness benefits through co-breeding (laying an egg in the same nest as the dominant female)[22,35]. Co-breeding subordinates always provide alloparental care and do not discriminate between their own or the dominant female's offspring (i.e. they help all offspring in the nest)[46,60]. Further, previous studies found no evidence for reproductive conflict caused by co-breeding females[35,40,61,62], except in extreme cases[32]. Therefore, we considered all subordinates that helped with incubation or provisioning as helpers, irrespective of whether they co-bred or not. Male subordinates acquire fewer benefits than females because they do not appear to benefit through gaining breeding experience[59] and very rarely gain direct paternity, which may explain why most helpers (88%;[36] 77% ($n = 310$) in this study) are female[35]. Apart from providing the opportunity to obtain indirect fitness benefits, the prolonged presence of the parents may be beneficial for subordinates because it facilitates the eventual acquisition of a dominant position elsewhere. This is because breeders are more likely to allow related subordinates to remain in the territory until they are able to disperse to a dominant position elsewhere, but will evict unrelated subordinates irrespective of such opportunities, resulting in higher mortality[33,63].

**Data collection.** For our analyses, we used data collected between 1995–2016, when the population was most intensively studied. We excluded the years 2000–2002 because fieldwork was limited in this period, with incomplete data on helping behaviour. In addition, we excluded 2004 because 58 individuals (both dominants and subordinates) were translocated to another island just before the main breeding season as part of a conservation programme[64], and 2005 because no territory quality data were collected in that year. During the main breeding season, each territory was monitored to determine the identity, helper status and number of group members and to assess breeding activity at least once every two weeks by following the resident dominant female for at least 15 min[55]. As the resighting probability for dominants during the main breeding season is virtually one[65], and migration is virtually non-existent[23], it is safe to assume that dominants not seen over an entire breeding season had died[25]. Once nest building commenced, each breeding attempt was monitored every 3–4 days until the nestling(s) fledged or the breeding attempt failed. To establish whether a subordinate provided nest care (helper) or not (non-helping subordinate) in a given season, we conducted nest watches of at least 60 min during both the incubation and nestling provisioning stages and recorded the start and end times of all provisioning events and incubation bouts and the identity of the individuals providing nest care[34]. For nests that failed early in the breeding stage (i.e. before an incubation and/or provisioning nest watch could be performed), subordinates were conservatively classified as non-helping subordinates. As in the majority of the territory-years where helpers were present (in 17% (271 out of 1571) of territory-years there was at least one helper present in the territory) there was only one helper of either sex (86% one helper, 14% two helpers, <1% three helpers; $n = 271$), we treated helper presence as a

binary variable (Y/N) in our analyses. Helper effects in cooperatively breeding species might result from factors associated with having subordinates (which are often retained offspring and thus indicate successful reproduction in previous years), such as differences in individual or territory quality, rather than from helping per se (see refs. [44,66]). Separating the impact of helping from individual or territory quality is extremely difficult, as experimentally manipulating the effect of help is generally not feasible or fraught with methodological issues[44]. Because in the Seychelles warbler not all subordinates help in any given year we can test statistically whether helper effects are better explained by having subordinates (i.e. living in a larger group), rather than by helping per se[34].

Seychelles warblers are almost entirely insectivorous, so we used an index of insect availability in each territory in each main breeding season as a proxy for territory quality (following refs. [54,67]). To calculate this, we used the formula $A * \sum (Cx * Ix)$, where $A$ is the size of the territory in ha, $Cx$ is the amount of foliage cover for tree species $x$, and $Ix$ is the mean monthly insect density for tree species $x$ per unit leaf area in $dm^2$. Territory size was determined from territory maps constructed from detailed observational data of foraging and territorial disputes. Foliage cover was determined by scoring the presence (i.e. >50% cover) or absence (i.e. ≤50% cover) of the 10 dominant tree species at the following height bands: 0–0.75 m, 0.75–2 m and each 2 m interval hereafter. This was done at 20 random points in each territory and the total number of presence scores, for each tree species, was our estimate of foliage cover. Insect densities were estimated by counting the number of insects on the undersides of 50 leaves for each of the 10 dominant tree species present at 14 different locations spread across the island. Insect counts taken at each location were used as an estimate for all territories near that location.

**Statistical analyses.** All models were performed separately for female and male dominants. Continuous predictor variables were centred and divided by two standard deviations to facilitate interpretation and comparison of model coefficients[68]. Non-significant ($P > 0.05$) interaction terms were removed, sequentially in order of least significance, from the models and final models contained all main effects and any significant interaction terms. We used R (version 3.2.5) for all analyses.

**Incubation attendance.** To investigate how dominant females respond to additional incubation performed by female helpers, we quantified incubation attendance for dominant females with and without female helpers. We predicted that dominant females would reduce their incubation attendance in response to being helped. For this, we used data on incubation behaviour that were collected between 2003 and 2015. For each nest, we calculated the dominant female's incubation attendance, which was the proportion of time the dominant female spent on incubation. In addition, we established whether only the dominant female incubated or whether there were additional incubators (helpers). We excluded all incubation observations of nests where the start or end time of one or more incubation bouts was unknown, because we were unable to calculate the incubation attendance in such cases. For the same reason, we also excluded observations of nests with female subordinates where the identity of the incubating individual could not be established for one or more incubation bouts. When multiple observations were performed at the same nest, only the first observation was selected. This resulted in 346 nest observations of 192 dominant females in 12 years. As incubation attendance approximated a normal distribution, we performed a linear mixed model (LMM) with Gaussian errors and an identity link function using lme4 (version 1.1-12[69]) in R. In this model, incubation attendance was the dependent variable and the fixed effects were log10 territory quality, the linear and quadratic effects of age (hereafter: age and $age^2$) of the dominant, helper presence (Y/N), the number of subordinates, and the two-way interactions between helper presence and the dominant's age and between the number of subordinates and the dominant's age. Dominant female identity and year were included as random effects. Subsequently, we repeated this analysis with the total incubation attendance by all incubating females (instead of the dominant female's incubation attendance) as the dependent variable to test the prediction that incubation by helpers leads to an increase in overall incubation attendance.

**Helping and actuarial senescence.** To investigate the impact of helping on age-dependent survival of dominants, we performed Generalized Linear Mixed Models (GLMMs) with a binomial error structure and a logit link function using the package lme4. Survival was a binary response variable stating whether a dominant survived until one year later than the season in which the breeding data were gathered[25]. Individual identity (which controls for repeated sampling of dominants and the territory they occupy throughout their breeding tenure[29]) and year (to control for unmeasured annual variation) were included as random effects. Models also included the following fixed effects: log10 territory quality, age and $age^2$ of the dominant, helper presence (Y/N), the number of subordinates, and the two-way interactions between helper presence and the dominant's age, and between the number of subordinates and the dominant's age. A significant interaction between helper presence and the dominant's age may suggest that helpers affect the pattern of age-dependent survival in dominants. We first treated helper presence as a binary variable (Y/N) in our analyses. Subsequently, as female helpers contribute

more to parental duties in the Seychelles warbler[37] and therefore may have a larger impact on the dominant's survival, we investigated whether a model that included the presence/absence of both female and male helpers separately explained the data better (by comparing the AICc values of both models) than a model with helper presence per se. The results of this model that included both male and female helper presence are reported in Supplementary Table 2.

Furthermore, as the fit of a quadratic age model could be largely determined by changes occurring during early-life, when the sample sizes are largest, this could potentially lead to misleading inferences about changes occurring during late life[70]. Therefore, we confirmed the late-life changes suggested by the models with a quadratic effect of age by comparing, using two proportion z-tests, dominant survival with and without helpers for dominants younger than seven years and for dominants older than six years, where six years is the onset of reproductive senescence in this species[25,45].

**Telomere attrition rate.** We tested whether dominants that received help show reduced telomere shortening using LMMs with a Gaussian error structure and an identity link function. Each year during the main breeding season, ca 25% of the adult population is caught using mist nets and blood samples are collected by brachial venipuncture[71,72]. Following the procedures described in detail elsewhere[29–31], we used qPCR to measure relative telomere length (RTL; the concentration of telomeric DNA relative to the concentration of the single-copy gene GAPDH) in blood samples collected from the same individual in two consecutive years. As avian erythrocytes are nucleated, this measure is effectively the RTL of the erythrocytes that comprise the great majority of blood cells. We then calculated ΔRTL as the difference between RTL in year $t$ and RTL in year $t + 1$ (i.e. one year later) within each individual and related ΔRTL to helper presence, with negative values indicating telomere shortening and positive values lengthening[31]. As there were only two ΔRTL measures available for female dominants with a male helper, only one ΔRTL measure for male dominants with a male helper, and because female helpers contribute more than male helpers do (see results), we focussed on comparing ΔRTL in dominants with a female helper to dominants without. The results of a model that included the presence of helpers (irrespective of the sex of the helper) were similar and are reported in Supplementary Table 3. As ΔRTL values may be greater in individuals with greater initial RTL (e.g. due to measurement error or 'regression to the mean'), we included an individual's initial RTL as a covariate to the models. Further, we included log10 territory quality, log10 age of the dominant, a binary variable (offspring produced Y/N) stating whether offspring were born in the territory in year $t$ that reached at least three months of age (as a measure of reproductive investment) and the number of subordinates (irrespective of their helping status) as predictors and included individual identity, year and birth year as random effects[31].

To test if dominants with helpers had better initial condition than individuals without helpers we compared RTL and the July (i.e. at the start of the breeding season) body mass of dominants with and without helpers using LMMs. For the models of telomere length, we included log10 of dominant age, log10 territory quality, helper presence (Y/N) and the number of subordinates as predictors and included individual identity, year and birth year as random effects. For the models of early-season body mass, we included helper presence, age, $age^2$, time of day [morning (0600–1000 h), midday (1000–1400 h), afternoon (1400–1900h)], log10 territory quality and tarsus length as predictors and included individual identity and year as random effects[71]. There were no differences in telomere length and July body mass between dominants that were helped or not (Supplementary Table 5; Supplementary Table 6).

**Age-dependent helper prevalence and subordinate reproduction.** To test the prediction that the probability that subordinates provide help increases among elderly dominants, we constructed GLMMs with a binomial error structure and a logit link function. Since the presence of helpers is conditional on subordinates being present in the territory, we tested these predictions on a dataset containing only dominants with one or more subordinates, with helping status of the subordinate (Y/N) as the dependent variable. First, we investigated the shape of the relationship between helping status and the dominant's age using generalized additive mixed models in the R package gamm4 02-4[73]. In these models, we fitted a non-parametric smoothing parameter for a dominant's age, which allows us to evaluate potential non-linear relationships between helper presence and a dominant's age[73]. As these models indicated a linear relationship between helper presence and age, we continued fitting age as a linear predictor in GLMMs. Age of the dominant, age of the subordinate (≤1 year old vs. ≥2 years old), sex of the subordinate, log10 territory quality and the number of subordinates in the territory were included as predictors. Dominant identity, family group, and year were included as random effects. We included an interaction between sex of the subordinate and the dominant's age to test whether the association between the dominant's age and the subordinate's likelihood of helping differed between male and female subordinates. To check if selective disappearance of poor-quality individuals could explain the age-dependent change in helping status, we added longevity of the dominant to the model (i.e. including only individuals that have died within our study period)[74]. As we found no evidence for selective disappearance effects (Supplementary Table 7), we report the results from the simpler models. Subsequently, we used a subset of the dataset containing only dominants

with helpers and tested whether the sex ratio among helpers changed with the age of the dominants. The sex of the subordinate was the dependent variable, age of the dominant, age of the subordinate, log10 territory quality, and the number of subordinates in the territory were included as fixed effects and dominant identity, family group, and year were included as random effects.

To test how the likelihood that subordinate females reproduced (co-breeding) was related to the age of the dominants, we constructed GLMMs with a binomial error structure and a logit link function. We used genetic parentage analyses based on 30 microsatellites using Masterbayes 2.52 to assign captured and genotyped offspring to subordinate females[75,76]. It should be noted that this is an underestimation of the total number of offspring that is produced as some offspring die before they can be captured and because we excluded offspring for which the genetic parents could not be assigned with at least 80% confidence[76]. Whether a subordinate female reproduced or not (Y/N) was the dependent variable and age of the dominant, age of the subordinate (≤1 year old vs. ≥2 years old), sex of the subordinate, log10 territory quality and the number of subordinates in the territory were included as predictors. Dominant identity, family group and year were included as random effects.

**Ethics statement.** The work was conducted with the permission of the Seychelles Bureau of Standards and the Seychelles Ministry of Environment, Energy and Climate Change and complied with all local ethical guidelines and regulations. Nature Seychelles provided permission to work on Cousin Island.

**Reporting summary.** Further information on experimental design is available in the Nature Research Reporting Summary linked to this article.

## Data availability

The data that support the findings of this study are available in figshare with the identifier https://doi.org/10.6084/m9.figshare.7751099.

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

## Acknowledgements

We thank Nature Seychelles for providing facilities to work on Cousin Island. The Seychelles warbler study would not have been possible without the help of many field-workers, lab technicians, database managers, students and researchers during the whole study period. The study was funded by Natural Environment Research Council (NERC) grants to DSR (NE/F02083X/1) and DSR and HLD (NE/K005502/1) and by Netherlands Organisation for Scientific Research (NWO) Grants 854.11.003 and 823.01.014 to J.K. (with D.S.R., T.B. and H.L.D.). H.L.D. was also funded by a NERC post-doctoral fellowship (NE/I021748/1). Both M.H. and S.A.K. were funded by NWO VENI Fellowships (863.15.020 and 863.13.017, respectively).

## Author contributions

M.H., D.S.R., J.K and S.A.K. designed the study. All authors performed research, including specifically; fieldwork – D.S.R., M.H., S.A.K., and K.B.; molecular work – K.B., L.S. and D.S.R. M.H. wrote the first draft of the manuscript. All authors provided input into concepts and ideas and critically revised the manuscript. M.H. analyzed the data with feedback from D.S.R and S.A.K. D.S.R., J.K., T.B. and H.L.D. coordinated the long-term study.
