## [Peer Review File · Nature Communications]

Reviewers' Comments:

Reviewer #1:

Remarks to the Author:

Review of 'Sociality delays senescence: evidence from a cooperatively breeding bird' by Hammers and co-authors

This study investigates whether the presence of helpers may decrease the rate of senescence and age-specific mortality using data from a long-term study on cooperatively breeding Seychelles warblers. The authors present convincing evidence of a link between the presence of helpers and delayed senescence. These are novel results that suggest a potentially important benefit of cooperation in this species that can be relevant to other species. This is, however, a correlative study, and hence causation cannot be established. But the authors are careful to point out the limitations of the correlative approach and avoid making unjustified claims. Overall, I found the study very interesting, the methods seemed adequate and the manuscript is well written. I do however have several comments that I believe deserve attention. I list these below.

Title

I believe that in the title you should not use the word 'sociality' but rather 'helpers', because in your study you find an effect of the presence of helpers, but not of group size, so sociality per se does not appear to have an effect on senescence, but the presence of helpers does

Intro

L40-42. Studies have also shown that the presence of helpers may be particularly beneficial for young and, presumably, inexperienced breeders (e.g. Magrath 2001 JAE, Paquet et al 2015 JAE). I know your paper is about the other end of life, but I think you should rephrase to give a more accurate picture

L45-54. The reasoning here makes sense, but I also had the impression there was some bias and a posteriori predictions. First, older birds might accumulate helpers simply because they've reproduced more and hence have more offspring, which means have more potential helpers? Additionally, although it makes sense to help older individuals if that makes more of a difference in terms of reproductive output, if younger breeders are nonetheless more productive than older breeders, and if there are younger relatives breeding, potential helpers might prefer to direct their help towards those younger, more productive relatives? You do not mention productivity in this study, but one is left thinking that it would be interesting to model that in order to better understand the impact of age-specific help on lifetime reproductive output. All this is to say I believe that you should not put so much emphasis on this part in the introduction.

Results

L79-81. It is strange that there are no overall differences among dominants with and without helpers, given such marked effects in late life. Are there differences in longevity or the frequency of age distribution among the two groups?

L99-102. Saying that you are 'excluding potentially confounding effects of group size in a different way' is a bit vague. Explain why you think it's important to repeat these analyses once more?

L108-110. I might be missing something here, but I don't see what fitting this three-way interaction is telling you. The previous test looking at presence of female or male helpers seems enough to me. Also, this procedure is not explained in the methods, and it should be

L115-119. As above, why testing the same thing with an interaction and separate analyses?

L146-151 I don't understand breaking this into dominant females and dominant males, because surely you follow the same breeders/groups, and hence your dominants should be more or less half females and half males and with the same number and sex ratio of helpers?? You say in lines 255-256 that 'Individuals that have acquired a dominant breeding position generally defend the same territory until their death'. So if in the majority of cases a pair stays together most of their lives, whether or not they have helpers and the sex ratio of those helpers should be similar for both? Or is the rate of divorce very high?... Please be clearer about this

Discussion

I like the way the authors are cautious about their findings and acknowledge possible confounding effects, which are always problematic in correlative studies such as this one. But I found that what is missing from this Discussion is the lack of overall differences in survival among dominants with and without helpers. This is strange given the marked effects of helpers on survival in late life, and I think this should be acknowledged and discussed. How do you think this result comes about and could the overall lack of difference between these two groups decrease the evolutionary significance of your age-related findings?

L159-160. Helper presence was associated with reduced telomere shortening in females, not in males. Please say so here, otherwise it can be misleading

L162. Rephrase to read 'our results indicate that female helpers contribute to delay senescence...'

L196-198. The causality problem is still there, as better quality females could still simultaneously have lower telomere shortening and more helpers because of higher intrinsic quality

L223-228. But you also need to explain the demographics of your couples (see my comment on the methods and results). We need to understand whether females have longer territory tenancy than males. Do they have higher survival or longevity and hence are there more chances of finding older dominant females than older dominant males, for example?

L234-241. In this study you also find that survival does not vary between dominants that received help vs those that didn't; it is in the older age that you find the benefits of helpers. This result need to be discussed, either here or somewhere else in the Discussion

Methods

It wasn't always clear to me which terms and interactions were tested in each model, and this should be clarified. I am also slightly concern about the breaking down of the statistical tests into several specific analyses and repeating analyses and this should be better justified or simplified by removing some of the analyses. I give specific examples below.

L307-310. L307-310. I wonder why you started by testing the effect of helpers as a binary variable and then repeated the analyses with the new variable male or female helper. Why didn't you use this variable from the beginning, since it seems more relevant given what you already know about the effects of helpers in this species?

L337-340. Did you first test for a sex effect? And why wasn't RTL included from the beginning, as it clearly seems to be an important factor? It is not clear to me exactly which terms you tested in these models – please clarify?

L350-351. As also stated in lines 146-151, I don't understand the logic of testing separately male and

female breeders, since in your study system the pairs seem to be quite stable. I would think it makes more sense to test whether the chances of having helpers increase with the couple's age. If this suggestion doesn't make sense, you should provide more information about why not. An alternative could be to look at the relationship with dominant females only, since you know they are the ones that benefit more from helper presence?

L341- 355. Please say exactly what was your dependent variable and the terms tested in the models? We get a bit lost as you start by describing your analyses of the relationships between the data.

L353-355. I also don't understand exactly what was tested here and how this provides a test of whether the sex ratio among helpers changed with the age of the dominants.

Reviewer #2:

Remarks to the Author:

This paper deals with an interesting topic (whether helping behavior mitigates senescence in cooperative breeders) and presents some promising findings from a valuable long-term data set that are suggestive of possible links between helping and aging. However, as the manuscript presents solely correlative data and several key analyses are underdeveloped, as multiple plausible alternative explanations would seem to exist for the key findings, and as compelling evidence would seem to be lacking for the key mechanism invoked to explain the findings, I'm afraid that I did not find the manuscript convincing. I hope the detailed comments below prove useful in highlighting points that could be usefully attended to.

MAJOR COMMENTS

The following points reduced my confidence in the three main analyses in the paper...

1. Helper effects on survival in old age

(a) There would appear to be several plausible alternative explanations that would not require a causal effect of helping.

As the evidence is purely correlative it would seem quite possible that the findings reflect higher quality dominants being both more likely to survive in old age and more likely to either have had reproductive success in the preceding years that has yielded the present helpers (as the helpers are likely to be offspring from previous years) or to be helped by the subordinates that they have. The findings could also be explained by variation in territory quality (with higher quality territories yielding both better survival in old age and helpers); this possibility cannot currently be ruled out, as while the survival analysis controlled for an index of insect availability (termed 'territory quality'), territories may vary in quality in many other ways that this variable alone does not capture. As previous work on this study population has reported that offspring are also more likely to delay dispersal on higher quality territories, and as subordinates that are delayed dispersers (and hence related to their dominants) are more likely to help than unrelated subordinates, the presence of helpers could also presumably correlate with territory quality via this mechanism. As such variation in individual quality or territory quality could explain both the survival results and the telomere results (see below), any apparent congruence between the two sets of findings would not seem to render the case for a helper effect any more compelling.

(b) Compelling evidence would seem to be lacking for the mechanism invoked by which helping could have such a causal effect.

The authors invoke helper effects on the work of dominants as the most likely mechanism that could

explain their findings. However, compelling evidence that this species shows helper-induced load-lightening appears to be lacking, both within this manuscript and in the published paper cited as providing such evidence. In the introduction of the manuscript, it is stated that "In response to being helped, dominants reduce their reproductive workload²⁸". This implies that causal evidence of such load lightening is provided in the cited paper. Inspection of the cited paper reveals that the presence of helpers actually had no detectable effect on the workload of dominant females (either their incubation effort or their provisioning rate) and the experimental removal of helpers had no effect on dominant survival. What evidence of load lightening there was was correlative and restricted to dominant males, whose provisioning rates were lower when helpers were present. As such, the cited published evidence of load-lightening in this species cannot readily explain the key results in this paper (i.e. why dominant females are apparently more likely to survive when helpers are present, why the telomeres of dominant females and not dominant males shorten less quickly when helpers are present, and why female helpers might have an effect on survival while male helpers do not). As this is a keystone piece of the logic, experimental evidence demonstrating such load-lightening would usefully strengthen the manuscript. Some unpublished material relating to load-lightening is alluded to in the discussion, but this is stated rather vaguely and requires a leap of faith.

(c) Might the link be causal because female 'helpers' actually disrupt dominant reproductive effort when they co-breed rather than because they 'help' to lighten it?

Given that female helpers routinely co-breed in this species, it would seem conceivable that associations between the presence of female helpers and the aging of dominants might arise in this species not because female helpers are helpful, but because female co-breeders increase the failure rate of the dominant's reproduction (e.g. see citation 28 for apparent evidence of this) and so actually lighten the total reproductive effort of the dominants over the breeding season. While the current emphasis is on cooperative 'helpers' having beneficial effects on dominants' lifespans, in this scenario conflict with co-breeding 'helpers' could actually be compromising reproductive investment among dominants, increasing their survival probability as a byproduct. Either way it would be good to see this alternative explanation considered.

In order to attribute any effects of female 'helpers' to the fact that they are providing alloparental care to the dominant's young (rather than co-breeding and potentially disrupting the dominant pair's reproductive effort), it would be nice to show that the effect of female helpers on dominant survival is apparent in years when those female helpers are simply helping and not co-breeding.

(d) Statistical evidence that female helpers have a greater effect on actuarial senescence than male helpers.

The results text currently implies that statistical evidence has been presented that female helpers have a greater effect on actuarial senescence than male helpers, but it was not clear to me that this was the case. Lines 108-111 would appear to misinterpret the meaning of this three way interaction, and inspection of the effect sizes in the next sentence would seem to reveal no clear helper-sex difference in the effect size estimate of either (1) the main effect of having a helper of that sex or (2) the interaction between having a helper of that sex and dominant-age (the standard errors around the effect sizes for having male and female helpers appear to be overlapping in both cases). A 3-way interaction of this kind implies to me that the effect of female helpers on age-related survival depends on whether there is a male helper present or not (as one might expect), rather than that the effects of male and female helpers differ as implied. On a separate note, I also wondered whether an effect of helpers on dominant survival could be less apparent here for male helpers than female helpers simply because the sample size of data points with male helpers present is likely to be much smaller than the sample size of data points with female helpers present (see below too).

2. Helper effects on telomere shortening.

This analysis seemed underdeveloped given the weight of information available about the individuals involved. The analysis would appear to indicate that when no other variables are controlled (apart from variation in the initial telomere length measurement) the rate of telomere attrition was lower among dominant females that had a female helper than dominant females that did not. As there are a host of plausible confounds of such a correlation, it is surprising that no attempt has been made to control for them... e.g. group size effects rather than helper effects, variation in the age of the dominant, variation in reproductive effort among the dominants in the focal year, variation in the quality of the dominant and of their territory, year effects etc. As outlined above, the case for a causal effect of helpers on telomere shortening in dominant females would also seem to be undermined by the lack of compelling evidence that helpers lighten the workloads of dominant females in this species.

3. Changes in helper prevalence with dominant age.

This also seemed a cursory analysis given the data available. The observation that older females with one subordinate are more likely to be helped by that subordinate is interesting. However, there would seem to be an array of potential explanations for this finding other than preferential helping of old females by subordinates, which could have been usefully explored with further analyses (e.g. confounding effects of the age of the subordinate or whether the subordinate was natal or an immigrant on their likelihood of helping, and the possibility that the dominant age effect reflects among-female differences coupled with a selective disappearance effect rather than a within-female effect of increasing in age).

It could also be usefully clarified how the analysis worked once attention was shifted to the effect of female helper presence versus male helper presence. Specifically, should we interpret the more apparent dominant female age effect on female helper presence as evidence that with increasing dominant female age (1) the one subordinate that those dominant females with one subordinate of either sex have is increasingly likely to be a female subordinate that helps (a relationship that could be a product of age-related changes in sex allocation for example), or that (2) the one subordinate that those dominant females with one subordinate female have is increasingly likely to help?

On a separate note, given the potential for female helpers to inherit the dominant breeding position on the death of their dominant female (presumably?), while it is suggested that female helpers stand to gain fitness benefits from helping their aging mothers to survive, it would be good to consider the possibility that female helpers could instead suffer a substantial direct fitness cost by helping their mothers to survive if doing so reduces their own chances of inheriting the local dominant position.

SPECIFIC COMMENTS

Introduction

The statement that "studies investigating the relationship between sociality and senescence at the intraspecific level are rare" could be usefully accompanied by acknowledgement of the studies that have looked at the impact of sociality on senescence at the intraspecific level in the context of competition rather than cooperation.

Results

The rarity of male helpers precluded the analysis of the impact of male helpers on changes in telomere length. As such, it would be useful to state the sample sizes for the other analyses with regard to the presence and absence of male helpers and the presence and absence of female helpers. For example,

I was left wondering whether there might be no apparent effect of male helpers on dominant survival principally because the sample size of data points with male helpers present is much smaller than the sample size of data points with female helpers present (see above).

Discussion

Line 162 - "our results indicate that that [sic] female helpers help to delay senescence in dominants" would seem to need some toning down given the potential for alternative explanations to account for the findings (see above).

Line 206-208 – speculation that helpers might also mitigate reproductive senescence among dominants could be usefully replaced with analyses establishing whether this is the case, given that such data are available.

Lines 214-215 – the manuscript would benefit throughout from a clearer logical distinction being drawn between co-breeding and helping, as it would seem that female 'helpers' may do one or the other in any given breeding attempt, leaving it unclear which behavior might be driving the relationships observed if they were indeed causal. Indeed, if helping is defined as the provision of alloparental care, and co-breeding in this species involves putting offspring in the dominant's nest, it would be good to clarify whether female co-breeders are actually really helpers at all in that particular breeding attempt.

Methods

Lines 337-338 – "We performed this analysis for male and female dominants separately, as these relationships differ between dominants of the two sexes (see results)." The results don't appear to show a significant effect of the sex of the dominant on the magnitude of the helper effect on change in telomere length. It would be good to show this, by testing for the 3 way interaction: dominant age x female helper presence x dominant sex. Likewise for the similar statement for the helper presence analysis (lines 350-352).

Reviewer #3:

Remarks to the Author:

This manuscript contains important new information from a classic long-term of Seychelles warblers. Among other results the authors present evidence that: (i) female but not male helpers enhance annual survival of the dominant female; (ii) dominant females but not males suffer telomere shortening if unhelped, but telomere lengthening if helped; and (iii) the likelihood that a subordinate helps (provisions at the nest) increases for female but not male helpers with the age of the dominant female but not the dominant male.

Taken alone, these results are important and likely to attract considerable interest. The paper is mostly clearly written and is very well-illustrated. It is remarkably free of typographical errors, and the data are for the most part appropriately analysed. It is also commendable in being based on a very substantial body of field work (albeit in a nice place).

Despite this, and my having read a reasonable proportion (though by no means all) of the Seychelles warbler papers, I found myself repeatedly asking 'but what if ...' and 'does this happen ...' questions, which make me feel that a very strong manuscript would become even better through some more definitive description of the helping system, and an elaboration and testing of the predictions associated with the models sketched in the second paragraph of p 3.

As I understand the system, the birds occupy territories, on which it is usually possible to identify a dominant pair, but there are also supernumerary birds. While the overwhelming majority of supernumerary males do not help, supernumerary females can (i) live on the territory without helping; (ii) help by feeding the offspring of the dominant female without direct contributions to reproduction, or (iii) co-breed by laying one of the eggs in what becomes a two-egg brood. It seems that the helping birds are often retained but on p. 3 mention is also made of attracting helpers, and the different expectations that then arise if the supernumerary birds are kin. I think the manuscript would be vastly improved if we knew the age structure, kinship to the dominant, and proportion and type of subordinates that have dispersed or remained philopatric, as these can provide further evidence of the authors' conceptual model.

For example, in Fig 4 there is evidence that old females but not old males are increasingly likely to have helpers (it is not clear - and should be - how co-breeders are treated here). This is consistent with the authors' conceptual model. However, if it is due to helpers benefiting from assisting kin to resist senescence it poses some expectations on the knowledge that the helpers have about the social milieu in which they live. Most importantly, there has to be a recognition by the potential helper that the mother is old. There would possibly be direct knowledge that this is true if the potential helper had been on the territory herself for many years, and would be supported by an increase in help by old helpers, but the relevant data are not given. However, relatedness to the male being assisted is likely to decline with age because the female changes her social or extra-pair partner. The second possibility is an increase in 'staying' incentives by old dominants. This presumably predicts an increase in co-breeding with the age of the mother, but the relevant data are not given, though again they are presumably available, and I suspect in this case there may be alternative explanations.

The other issue is the untested notion that a female queuing for dominance gains a lifetime fitness benefit from prolonging the life of the bird at the top of the queue, and hence delaying their eventual succession. More information on the pattern of succession would be helpful, but I suspect some modelling would be necessary here.

Minor comments

Why are model predictions illustrated in this Figure, but raw means in the other Figures

It is uncertain what is meant by the sample sizes in Figure 4. A single sample size implies that each female has both a male and female subordinate, which seems unlikely to be true.

We thank the three reviewers for their thorough, constructive and insightful comments. We have considerably revised the manuscript and analyses in light of these comments, and performed additional analyses in order to address the concerns raised by the reviewers. We feel that these revisions have improved our manuscript considerably.

We responded to each point raised by the referees. We have numbered the reviewer comments to allow for easy cross-referencing.

Reviewers' comments:

Reviewer #1 (Remarks to the Author):

Review of 'Sociality delays senescence: evidence from a cooperatively breeding bird' by Hammers and co-authors

This study investigates whether the presence of helpers may decrease the rate of senescence and age-specific mortality using data from a long-term study on cooperatively breeding Seychelles warblers. The authors present convincing evidence of a link between the presence of helpers and delayed senescence. These are novel results that suggest a potentially important benefit of cooperation in this species that can be relevant to other species. This is, however, a correlative study, and hence causation cannot be established. But the authors are careful to point out the limitations of the correlative approach and avoid making unjustified claims. Overall, I found the study very interesting, the methods seemed adequate and the manuscript is well written.

We are pleased that the reviewer is positive about our manuscript and thank the reviewer for these constructive comments.

I do however have several comments that I believe deserve attention. I list these below.

Title

1. I believe that in the title you should not use the word 'sociality' but rather 'helpers', because in your study you find an effect of the presence of helpers, but not of group size, so sociality per se does not appear to have an effect on senescence, but the presence of helpers does

We agree with the reviewer and have replaced "sociality" with "helpers" in the title.

The title now reads: "Helpers delay parental senescence: evidence from a cooperatively breeding bird"

Intro

2. L40-42. Studies have also shown that the presence of helpers may be particularly beneficial for young and, presumably, inexperienced breeders (e.g. Magrath 2001 JAE, Paquet et al 2015 JAE). I know your paper is about the other end of life, but I think you should rephrase to give a more accurate picture

We have rephrased this sentence to acknowledge that not only elderly dominants may benefit from having helpers, but also that young, inexperienced dominants should benefit because they have little breeding experience. We have included both suggested references to support this statement.

3. L45-54. The reasoning here makes sense, but I also had the impression there was some bias and a posteriori predictions. First, older birds might accumulate helpers simply because they've reproduced more and hence have more offspring, which means have more potential helpers? Additionally, although it makes sense to help older individuals if that makes more of a difference in terms of reproductive output, if younger breeders are nonetheless more productive than older breeders, and if there are younger relatives breeding, potential helpers might prefer to direct their help towards those younger, more productive relatives? You do not mention productivity in this study, but one is left thinking that it would be interesting to model that in order to better understand the impact of age-specific help on lifetime reproductive output. All this is to say I believe that you should not put so much emphasis on this part in the introduction.

We agree that we had put too much emphasis on the potential mechanism underlying this prediction in the introduction. Therefore, we have considerably shortened and simplified this section in the introduction. This section now reads:

(L49-51): “If the benefits of receiving help increase with a dominant’s age, there should be strong selection on elderly dominants to recruit and retain helpers. Therefore, we predict that the likelihood of having helpers increases with age in dominants.”

Testing the impact of help on productivity is outside the scope of this manuscript, but we do discuss the possibility of age-dependent accumulation of helpers due to higher past reproduction (L184-188) and mention the possibility that subordinates might more likely to help when they assess the dominants as being in better physiological condition or more productive (L194-196). We also discuss the age-dependent increase in helper prevalence and the potential impact of helpers on age-specific productivity in detail in the discussion (L197-228).

Results

4. L79-81. It is strange that there are no overall differences among dominants with and without helpers, given such marked effects in late life. Are there differences in longevity or the frequency of age distribution among the two groups?

The absence of an overall difference in survival is probably because there is no detectable difference in survival for younger and mid-aged dominants that received help or not. Only a small proportion of individuals reach the age at which the survival differences become apparent (see Fig. 1), so this comparison is mostly influenced by the younger age groups. We now include a test of the difference in survival with and without a female helper for younger (<7 years) and older (>6 years) dominants and confirm that helpers only affected the survival of elderly dominants (L81-86).

We investigated annual helper presence (and thus the presence of helpers differed across years for the same individual), so were not able to investigate the impact of helper presence on dominant longevity. Testing the impact of helpers on longevity would require a lifetime measure of the total help received by a dominant, but this is problematic as such a measure is likely to be a function of longevity itself.

We have investigated the age distribution of individuals with and without helpers and added these figures below and in the supplementary material (Figure S1, see below). The age distributions of male dominants with and without male or female helpers appear similar, but female dominants with female helpers are often older than those with only male or no helpers. This pattern is probably due to the tendency for female subordinates to remain in the territory as a helper when their presumed mother (i.e. the dominant female) is still present (Komdeur et al. 2004¹).

Figure S1: Age distribution of female and male dominants. A: irrespective of helper presence; B: without helpers; C: with female helpers; D: with male helpers.

5. L99-102. Saying that you are 'excluding potentially confounding effects of group size in a different way' is a bit vague. Explain why you think it's important to repeat these analyses once more?

We agree that this overlaps with our other analysis of group size effects, so we have now omitted it. We now present just one analysis, which includes group size and helper presence in the same model. The benefit of doing this is that, because not all subordinates provide help, we are able to separate the effect of helping from the impact of living in a larger group.

6. L108-110. I might be missing something here, but I don't see what fitting this three-way interaction is telling you. The previous test looking at presence of female or male helpers seems enough to me. Also, this procedure is not explained in the methods, and it should be

We agree with the reviewer that testing this three-way interaction is uninformative and we have now removed it.

7. L115-119. As above, why testing the same thing with an interaction and separate analyses?

As we now present separate analyses for male and female dominants and do not statistically test whether the patterns differ between male and female dominants, this comment is no longer applicable.

8. L146-151 I don't understand breaking this into dominant females and dominant males, because surely you follow the same breeders/groups, and hence your dominants should be more or less half females and half males and with the same number and sex ratio of helpers?? You say in lines 255-256 that 'Individuals that have acquired a dominant breeding position generally defend the same territory until their death'. So if in the majority of cases a pair stays together most of their lives, whether or not they have helpers and the sex ratio of those helpers should be similar for both? Or is the rate of divorce very high?... Please be clearer about this

The reviewer is correct that there are approximately similar numbers of dominant males and females but the dynamic nature of pair formation – where dominants that die are replaced by younger birds – means that there is a relatively weak correlation between ages within a pair ($r = 0.16$, $t_{1531} = 6.53$, $P < 0.001$, see figure S3, which is also shown below). This allows us to evaluate the impact of age of male and female dominants separately. We have now clarified this in the text.

(L255-260): “The correlation between the age of the dominant male and female in a territory is, while significant, actually relatively weak ($r = 0.16$, $t_{1531} = 6.53$, $P < 0.001$, Fig. S3). This is because the age at which an individual obtains a dominant position varies considerably, pairs of birds do not become dominant at the same age, and the age at which dominant individuals die (and one side of the dominant pair is replaced) varies.”

Figure S3: The ages of the male and the female dominant within a pair. The $x = y$ slope is given for reference. The size of the data points is proportional to the number of occurrences of each combination of male and female age. Numbers are sample sizes.

Discussion

9. I like the way the authors are cautious about their findings and acknowledge possible confounding effects, which are always problematic in correlative studies such as this one. But I found that what is missing from this Discussion is the lack of overall differences in survival among dominants with and without helpers. This is strange given the marked effects of helpers on survival in late life, and I think this should be acknowledged and discussed. How do you think this result comes about and could the overall lack of difference between these two groups decrease the evolutionary significance of your age-related findings?

See also our response to comment 4, where we point out that we inevitably have relatively fewer data for the older age classes. Furthermore, the very high annual survival probability in young and mid-aged individuals means there is little scope for improvement in survival (e.g. a ‘ceiling effect’), whereas there is much more potential for this in elderly individuals with lower survival probabilities. We have now discussed this and the evolutionary implications:

(L152-156): “The finding that only elderly individuals, that normally have lower survival because of senescence, benefit from receiving help, could be caused by a “ceiling effect”: the very high annual

survival in young and mid-aged individuals means there is little potential for improvement in survival, but there is much more scope for this in elderly individuals with lower survival probabilities.”

(L236-242): “Further, because the force of natural selection is proportional to the number of individuals alive in a given age class (Hamilton 1966), the small number of elderly dominants that benefit from help (Figure 1) means that selection on delayed senescence may be relatively weak compared to factors that improve fitness in early life. However, a positive effect of helpers on the dominant’s fitness in late life should nonetheless select for delayed senescence and longer lifespan in dominants, and increased cooperative breeding.”

10. L159-160. Helper presence was associated with reduced telomere shortening in females, not in males. Please say so here, otherwise it can be misleading

We agree, and so have added “in dominant females, but not in dominant males” to this statement.

11. L162. Rephrase to read 'our results suggest that female helpers contribute to delay senescence...'

Done

12. L196-198. The causality problem is still there, as better quality females could still simultaneously have lower telomere shortening and more helpers because of higher intrinsic quality

Because we did not perform an experiment, we are not able to show causality here. Therefore, we have now been more careful with our wording here. In addition, we now test whether individuals with a helper were initially in better condition and had longer telomeres (i.e. less previous telomere shortening) and higher body mass. Female and male dominants with or without helpers did not differ in initial telomere length and pre-breeding body mass, which are known indicators of condition in this species, which suggests that those with helpers were not of initial better quality. We have added these analyses to the methods and to the supplementary material.

Specifically, we have added the following:

(L367-377): “To test if dominants with helpers had better initial condition (were of better quality) than individuals with helpers had we compared RTL and July (i.e. at the start of the breeding season) body mass of dominants with and without helpers using LMMS. For the models of telomere length, we included log₁₀ of dominant age, territory quality, helper presence (Y/N) and the number of subordinates as predictors and included individual identity, year and birth year as random effects. For the models of early-season body mass, we included helper presence, age, age², time of day [morning (0600-1000h), midday (1000-1400h), afternoon (1400h-1900h)], territory quality and tarsus length as predictors and included individual identity and year as random effects (Kingma et al. 2016). There were no differences in telomere length and July body mass between dominants that were helped or not (Table S5, Table S6). Repeating the model with female helper presence (Y/N) instead of the presence of a helper of either sex gave identical results (not shown).”

In L188-192 we now state the following: “Moreover, the greater telomere shortening (a longitudinal measure across two points within each female) observed in female dominants that were not helped, compared to the lack of shortening in helped females, suggests that helpers prevent a deterioration of the dominant female’s condition, rather than that dominants with a helper were initially in better condition (of better quality).”

13. L223-228. But you also need to explain the demographics of your couples (see my comment on the methods and results). We need to understand whether females have longer territory tenancy than males. Do they have higher survival or longevity and hence are there more chances of finding older dominant females than older dominant males, for example?

Previous studies on the Seychelles warbler showed that male and female dominants have similar breeding tenure, annual survival probabilities and rates of actuarial senescence (Brouwer et al. 2006², Hammers et

al. 2013³). Please also see our response to comment 7 regarding the age-distribution of couples, where we have provided a figure showing this age-distribution.

We have added this to L260-261: “Previous studies have shown that male and female dominants have similar breeding tenure, annual survival probabilities and rates of actuarial senescence^{2,3}.”

15. L234-241. In this study you also find that survival does not vary between dominants that received help vs those that didn't; it is in the older age that you find the benefits of helpers. This result need to be discussed, either here or somewhere else in the Discussion

This comment is similar to comment 9 and we have now discussed this in the discussion. See our response to comment 9.

Methods

16. It wasn't always clear to me which terms and interactions were tested in each model, and this should be clarified. I am also slightly concern about the breaking down of the statistical tests into several specific analyses and repeating analyses and this should be better justified or simplified by removing some of the analyses. I give specific examples below.

We have carefully gone through the methods section and have made several clarifications regarding the specifications of the models. In addition, we have added the full tables for all model results, so that it is immediately clear which terms and interactions were tested in each model. We have also removed several ‘repeat’ analysis that did basically test the same thing (see responses above to comments 5-7). Below, we respond to the specific examples.

17. L307-310. L307-310. I wonder why you started by testing the effect of helpers as a binary variable and then repeated the analyses with the new variable male or female helper. Why didn't you use this variable from the beginning, since it seems more relevant given what you already know about the effects of helpers in this species?

We chose this approach because we first aimed to establish whether survival was associated with the presence of helpers (as both male and female helpers contribute to offspring provisioning) and then set out to partition this into contributions by female and male helpers as their contributions differ (female helpers generally contribute more than male helpers: female helpers contribute more to offspring provisioning, and female helpers, unlike male helpers, contribute to nest building and incubation). We have explained this more clearly in the Methods.

18. L337-340. Did you first test for a sex effect? And why wasn't RTL included from the beginning, as it clearly seems to be an important factor? It is not clear to me exactly which terms you tested in these models – please clarify?

We now provide separate analyses for male and female dominants for all analyses and do not explicitly test for a sex difference, so this comment is no longer applicable.

We have now included initial telomere length from the beginning and more clearly explained in the methods which terms were tested. In addition, for all main analyses we now provide tables with full model outputs.

19. L350-351. As also stated in lines 146-151, I don't understand the logic of testing separately male and female breeders, since in your study system the pairs seem to be quite stable. I would think it makes more sense to test whether the chances of having helpers increase with the couple's age. If this suggestion doesn't make sense, you should provide more information about why not. An alternative could be to look at the relationship with dominant females only, since you know they are the ones that benefit more from helper presence?

See our response to comment 7 (and the figure shown there); because breeders that have died are replaced by (often younger) new breeders, there is enough variation in the ages of male and female

dominants in our dataset to assess this relationship for males and females separately.

20. L341- 355. Please say exactly what was your dependent variable and the terms tested in the models? We get a bit lost as you start by describing your analyses of the relationships between the data.

We have gone in detail through the methods section and better clarified which terms were tested in each model. We have also included tables for all main analyses, so that it is immediately clear which variables were included in each model.

21. L353-355. I also don't understand exactly what was tested here and how this provides a test of whether the sex ratio among helpers changed with the age of the dominants.

We have now clarified that we tested if, given there is a helper present, the sex of the helper changed with the age of the dominant.

(L378-401): “To test the prediction that the probability that subordinates provide help increases among elderly dominants, we constructed GLMMs with a binomial error structure and a logit link function. Since the presence of helpers is conditional on subordinates being present in the territory, we tested these predictions on a dataset containing only dominants with one or more subordinates, with helping status of the subordinate (Y/N) as the dependent variable. First, we investigated the shape of the relationship between helping status and the dominant's age using generalized additive mixed models in the R package *gamm4* 02-4⁴. In these models, we fitted a non-parametric smoothing parameter for a dominant's age, which allows us to evaluate potential non-linear relationships between helper presence and a dominant's age⁴. As these models indicated a linear relationship between helper presence and age, we continued fitting age as a linear predictor in GLMMs. Age of the dominant, age of the subordinate (≤ 1 year old vs. 2+ years old), sex of the subordinate, territory quality and the number of subordinates in the territory were included as predictors. Dominant identity, family group, and year were included as random effects. We included an interaction between sex of the subordinate and the dominant's age to test whether the association between the dominant's age and the subordinate's likelihood of helping differed between male and female subordinates. To check if selective disappearance of poor-quality individuals could explain the age-dependent change in helping status, we added longevity of the dominant to the model (i.e. including only individuals that have died within our study period), following Van de Pol & Verhulst⁵. As we found no evidence for selective disappearance effects (Table S7), we report the results from the simpler models. Subsequently, we used a subset of the dataset containing only dominants with helpers and tested whether the sex ratio among helpers changed with the age of the dominants. The sex of the subordinate was the dependent variable, age of the dominant, age of the subordinate, territory quality, and the number of subordinates in the territory were included as fixed effects and dominant identity, family group, and year were included as random effects.”

Reviewer #2 (Remarks to the Author):

This paper deals with an interesting topic (whether helping behavior mitigates senescence in cooperative breeders) and presents some promising findings from a valuable long-term data set that are suggestive of possible links between helping and aging. However, as the manuscript presents solely correlative data and several key analyses are underdeveloped, as multiple plausible alternative explanations would seem to exist for the key findings, and as compelling evidence would seem to be lacking for the key mechanism invoked to explain the findings, I'm afraid that I did not find the manuscript convincing. I hope the detailed comments below prove useful in highlighting points that could be usefully attended to.

We thank the reviewer for these insightful and very useful comments. We have taken the concerns of the reviewer to heart and (i) provide new evidence for load-lightening, (ii) discuss potential alternative explanations for our findings and (iii) explain more clearly why it is not possible to perform experimental manipulations of helping in this species. However, given these limitations and the positive effects of helpers on load-lightening for breeders (van Boheemen et al. *in press*⁶ and Supplement 1, spanning the same time period as our study on helpers and senescence), the evidence provided here is novel and, we believe, important – providing useful insight that will provoke new studies and analyses. We hope that our clarifications and new analyses have dealt with the issues raised about the analyses.

MAJOR COMMENTS

The following points reduced my confidence in the three main analyses in the paper...

1. Helper effects on survival in old age

(a) There would appear to be several plausible alternative explanations that would not require a causal effect of helping.

As the evidence is purely correlative it would seem quite possible that the findings reflect higher quality dominants being both more likely to survive in old age and more likely to either have had reproductive success in the preceding years that has yielded the present helpers (as the helpers are likely to be offspring from previous years) or to be helped by the subordinates that they have. The findings could also be explained by variation in territory quality (with higher quality territories yielding both better survival in old age and helpers); this possibility cannot currently be ruled out, as while the survival analysis controlled for an index of insect availability (termed 'territory quality'), territories may vary in quality in many other ways that this variable alone does not capture. As previous work on this study population has reported that offspring are also more likely to delay dispersal on higher quality territories, and as subordinates that are delayed dispersers (and hence related to their dominants) are more likely to help than unrelated subordinates, the presence of helpers could also presumably correlate with territory quality via this mechanism. As such variation in individual quality or territory quality could explain both the survival results and the telomere results (see below), any apparent congruence between the two sets of findings would not seem to render the case for a helper effect any more compelling.

Although we agree with the reviewer that it is (as in almost all studies on cooperative breeding vertebrates) not possible to establish with certainty whether the observed relationships are causal, we do not believe that the correlative nature of our study prevents it being novel and important. In fact, our finding that the age x number of subordinates (including non-helpers) interaction was not significant, whereas the interaction between age x helper presence was, suggests that the helping, rather than the presence of subordinates *per se*, explains the effects found in our study. Furthermore, we have included insect availability (a good proxy for territory quality – and exactly the same measure of territory quality the reviewer refers to) in our analyses and found that dominant female survival is higher in territories with higher food availability. We also found that male survival, telomere shortening rate and the likelihood that a subordinate helped was unrelated to insect availability (territory quality).

In our opinion, and this is also highlighted by the first and third reviewer, we have clearly acknowledged and discussed the limitations of our study and do not make unjustified claims about the causality of the patterns we find. However, we have taken great care to investigate the potentially confounding effects of

group size, territory quality and individual quality, and have carefully discussed the limitations and alternative explanations of our results in the discussion.

Separating the impact of helping from the effects of individual or territory quality is extremely difficult in studies on cooperatively breeding vertebrates as experimentally manipulating the effect of help is generally a flawed approach, because this disrupts group composition (see e.g. Cockburn et al. 2008⁷). Cockburn et al. (2008) discuss several correlative methods to deal with this issue and state that a method similar to the one we utilized in our study to separate the impact of helping from group size is “potentially powerful”. Separating group size from helping is usually impossible in other systems because all subordinate group members help. In the Seychelles warbler not all subordinates help, which makes this a very good system in which to separate the impacts of help from group size *per se*.

Regarding the impact of territory quality on delayed dispersal and helping, the reviewer is correct that this was observed in the early years of the Seychelles warbler study, but this effect has disappeared during later years of the study. Natal territory quality was not related to dispersal during the time period considered in this study (Eikenaar et al. 2007⁸, 2010⁹), probably because the among-territory variation in territory quality reduced drastically (Eikenaar et al. 2010) when the island-wide vegetation cover increased as a result of conservation interventions (see Komdeur & Pels 2005¹⁰).

As stated in Eikenaar et al. 2010:

“Using data collected in the Cousin Island population from 1985 to 1994, Komdeur (1996) showed that the higher the quality (in terms of food availability) of the natal territory, the longer independent offspring delayed dispersal. Because females were usually born on higher quality territories than males (Komdeur et al., 1997), females delayed dispersal longer than males (Komdeur, 1996). However, a study in the same population between 1995 and 2005 revealed that during this period males were just as likely to delay dispersal in their first year of life as were females (Eikenaar et al., 2007).”

The female bias in delayed dispersal in the Cousin Island Seychelles warbler population observed in the early stages of the study was absent in later years. The change in dispersal coincided with a dramatic reduction in the extent of among territory variation in quality on Cousin.

Furthermore, the quality of the natal territory no longer affected dispersal decisions of yearlings (Eikenaar et al., 2007).”

However, we also realise that we did not explain very well why we had performed additional analyses to separate the impact of helper presence from group size. The additional analyses suggest that the impact of helpers on the late-life survival of dominants was not simply due to differences in individual or territory quality. We performed those analyses to control for the fact that, because helpers are often retained offspring from previous breeding attempts, dominants living in larger groups (or having helpers) may be of higher quality or live on better quality territories. The finding that the associations between helper presence and dominant survival are not explained by (i.e. are independent of) the total number of helping and non-helping subordinates, suggests that the patterns are driven by the presence of helpers, and not simply by higher reproduction by higher-quality individuals or individuals living in higher-quality territories. We also found that female and male dominants with or without helpers did not differ in initial telomere length or pre-breeding body mass. This suggests that individuals with and without helpers were in similar condition and did not differ in overall quality, at least as measured by telomere length and body condition (see our response to comment 12 of reviewer 1). Finally, territory quality, which is a combined measure of insect availability and territory size, was included in all analyses. We have clarified all this in the methods section.

(b) Compelling evidence would seem to be lacking for the mechanism invoked by which helping could have such a causal effect.

The authors invoke helper effects on the work of dominants as the most likely mechanism that could explain their findings. However, compelling evidence that this species shows helper-induced load-lightening appears to be lacking, both within this manuscript and in the published paper cited as providing such evidence. In the

introduction of the manuscript, it is stated that “In response to being helped, dominants reduce their reproductive workload28”. This implies that causal evidence of such load lightening is provided in the cited paper. Inspection of the cited paper reveals that the presence of helpers actually had no detectable effect on the workload of dominant females (either their incubation effort or their provisioning rate) and the experimental removal of helpers had no effect on dominant survival. What evidence of load lightening there was correlative and restricted to dominant males, whose provisioning rates were lower when helpers were present. As such, the cited published evidence of load-lightening in this species cannot readily explain the key results in this paper (i.e. why dominant females are apparently more likely to survive when helpers are present, why the telomeres of dominant females and not dominant males shorten less quickly when helpers are present, and why female helpers might have an effect on survival while male helpers do not). As this is a keystone piece of the logic, experimental evidence demonstrating such load-lightening would usefully strengthen the manuscript. Some unpublished material relating to load-lightening is alluded to in the discussion, but this is stated rather vaguely and requires a leap of faith.

We agree with the reviewer that the reference cited here as evidence for load-lightening in the Seychelles warbler only partly supports load-lightening in this species. However, we have now included compelling evidence that load-lightening occurs in the Seychelles warbler, both with respect to a significant reduction in incubation by dominant females in response to incubation by a female helper (Supplement 1; see figure and table below), and a significant reduction in provisioning rates in dominants of both sexes in response to helping by a subordinate of either sex (van Boheemen et al. *in press*).

The analysis of female incubation attendance shows that female dominants have 19% lower incubation attendance when a female helper is present. We did not observe load-lightening for male dominants because male Seychelles warblers do not incubate. A 19% reduction in incubation attendance is a very substantial reduction, because this time is released for foraging by the dominant. This incubation load-lightening effect may explain why we did find an effect on telomere shortening rate of being helped by a female helper for female dominants, but not for male dominants.

Figure Supplement 1. Incubation attendance by dominant females in relation to helper presence. Boxplots show median and 5%, 10%, 25%, 75%, 90% and 95% quantiles. Numbers are sample sizes.

Table Supplement 1. Incubation attendance by dominant females in relation to helper presence.

Incubation effort dominant female	Estimate	SE	t	P
Intercept	0.50	0.02	31.26	<0.001
Female helper (Y/N)	-0.09	0.02	-5.20	<0.001
Random	Variance			
Individual ID	<0.01		366 observations	
Year	<0.01		198 individuals	
Residual	0.02		13 years	

Furthermore, we now refer to a recently accepted manuscript (in *Ecology and Evolution*) – that used long-term data that spanned the same period as this study – about the load-lightening benefits of helpers on the dominants’ provisioning rates (van Boheemen et al. *in press*). This manuscript shows that provisioning rates of dominants of both sexes with a helper are 5–13% lower than those of individuals without helpers. A preprint of van Boheemen et al. is available on [biorxiv.org](https://www.biorxiv.org/content/early/2018/07/19/372722): <https://www.biorxiv.org/content/early/2018/07/19/372722>

(c) Might the link be causal because female ‘helpers’ actually disrupt dominant reproductive effort when they co-breed rather than because they ‘help’ to lighten it?

Given that female helpers routinely co-breed in this species, it would seem conceivable that associations between the presence of female helpers and the aging of dominants might arise in this species not because female helpers are helpful, but because female co-breeders increase the failure rate of the dominant’s reproduction (e.g. see citation 28 for apparent evidence of this) and so actually lighten the total reproductive effort of the dominants over the breeding season. While the current emphasis is on cooperative ‘helpers’ having beneficial effects on dominants’ lifespans, in this scenario conflict with co-breeding ‘helpers’ could actually be compromising reproductive investment among dominants, increasing their survival probability as a byproduct. Either way it would be good to see this alternative explanation considered. In order to attribute any effects of female ‘helpers’ to the fact that they are providing alloparental care to the dominant’s young (rather than co-breeding and potentially disrupting the dominant pair’s reproductive effort), it would be nice to show that the effect of female helpers on dominant survival is apparent in years when those female helpers are simply helping and not co-breeding.

We thank the reviewer for this suggestion. However, the available evidence suggests that helpers improve the dominant’s reproductive success, rather than disrupting it (Richardson et al. 2002¹¹, Komdeur & Richardson 2007¹², Kingma et al. 2018¹³).

Komdeur (1994¹⁴) found that, in extreme cases with many helpers (i.e. 3–4 helpers), helpers may increase reproductive failure. However, this was only the case for nests with three or more helpers, whereas having one or two helpers has a positive effect. In our study, in <1% nests there were three helpers present and nests with more than three helpers did not occur (L299-300). The vast majority of nests with helpers had just one (86%) or two (14%) helpers, for which the effects on reproduction were positive. Further, more recent studies have found no evidence for reproductive conflict between co-breeders and dominant females (Richardson et al. 2002¹¹, Bebbington et al. 2017¹⁵, Kingma et al. 2018¹³). Instead, these studies show that co-breeders generally have a positive effect on the dominant’s reproductive success. Nests with co-breeding females suffer significantly less nest predation and have a much higher hatching success through higher incubation attendance (Kingma et al. 2018), and dominant females with a co-breeding helper produce more fledglings than do dominant females without co-breeding females (Richardson et al. 2002). Further, *per capita* (per nestling) provisioning rates in nests with co-breeding helpers are similar to those for nests with non-breeding helpers, and the body masses and telomere

lengths of nestlings do not differ between nests with co-breeding and those with nonbreeding helpers (Bebbington et al. 2017). Furthermore, the likelihood of co-breeding is not associated with territory quality (Bebbington et al. 2017).

We have added the following to the methods section:

L277-278: “Further, previous studies found no evidence for reproductive conflict caused by co-breeding females (Richardson et al. 2002, Komdeur & Richardson 2007, Bebbington et al. 2017, Kingma et al. 2018), except in extreme cases (Komdeur 1994).”

(d) Statistical evidence that female helpers have a greater effect on actuarial senescence than male helpers. The results text currently implies that statistical evidence has been presented that female helpers have a greater effect on actuarial senescence than male helpers, but it was not clear to me that this was the case. Lines 108-111 would appear to misinterpret the meaning of this three way interaction, and inspection of the effect sizes in the next sentence would seem to reveal no clear helper-sex difference in the effect size estimate of either (1) the main effect of having a helper of that sex or (2) the interaction between having a helper of that sex and dominant-age (the standard errors around the effect sizes for having male and female helpers appear to be overlapping in both cases). A 3-way interaction of this kind implies to me that the effect of female helpers on age-related survival depends on whether there is a male helper present or not (as one might expect), rather than that the effects of male and female helpers differ as implied. On a separate note, I also wondered whether an effect of helpers on dominant survival could be less apparent here for male helpers than female helpers simply because the sample size of data points with male helpers present is likely to be much smaller than the sample size of data points with female helpers present (see below too).

The reviewer is correct that this was a wrong interpretation on our part and we thank the reviewer for pointing this out. We have removed this three-way interaction and, because of the small sample size of male helpers and the skewed distribution of male helpers in relation to the dominants’ ages (few elderly dominants have male helpers, see Fig. 3, Fig. S1, Fig. S2), we do not now attempt to test if the age-dependent effect differs between male and female helpers. Indeed, the lack of an impact of male helpers could indeed be a sample size effect, as suggested by the reviewer.

We now explicitly state this in the results (L93-96) “We did not find an association between dominant female (age-dependent) survival and male helper presence (Table 1), though the likelihood of detecting such an effect is reduced because male helpers are much less common than female helpers, especially among elderly dominants (Supplementary Fig. 1, Supplementary Fig. 2).”

We have also removed our statement about the different contributions of male and female helpers to the age-dependent survival of dominants.

2. Helper effects on telomere shortening.

This analysis seemed underdeveloped given the weight of information available about the individuals involved. The analysis would appear to indicate that when no other variables are controlled (apart from variation in the initial telomere length measurement) the rate of telomere attrition was lower among dominant females that had a female helper than dominant females that did not. As there are a host of plausible confounds of such a correlation, it is surprising that no attempt has been made to control for them... e.g. group size effects rather than helper effects, variation in the age of the dominant, variation in reproductive effort among the dominants in the focal year, variation in the quality of the dominant and of their territory, year effects etc. As outlined above, the case for a causal effect of helpers on telomere shortening in dominant females would also seem to be undermined by the lack of compelling evidence that helpers lighten the workloads of dominant females in this species.

We agree. Indeed, we could have further developed this analysis to control for possible confounds and have done this now (L360-366). We have now included initial telomere length, territory quality, log₁₀ age of the dominant (see Spurgin et al. 2018), the number of offspring born in the territory in year *t* that reached at least three months of age (as a measure of reproductive investment) and the number of subordinates (irrespective of their helping status) as predictors, and have added year and birth cohort (see Spurgin et al. 2018) as random effects. As with the survival analyses, we checked whether our results

were explained by the number of subordinates in the group, rather than by helping, by including them in the same model. The results of this new analysis are the same and we have now included these results (L108-115, Table 2).

3. Changes in helper prevalence with dominant age.

This also seemed a cursory analysis given the data available. The observation that older females with one subordinate are more likely to be helped by that subordinate is interesting. However, there would seem to be an array of potential explanations for this finding other than preferential helping of old females by subordinates, which could have been usefully explored with further analyses (e.g. confounding effects of the age of the subordinate or whether the subordinate was natal or an immigrant on their likelihood of helping, and the possibility that the dominant age effect reflects among-female differences coupled with a selective disappearance effect rather than a within-female effect of increasing in age).

We have further developed this analysis and now also included the age of the subordinate (older subordinates are more likely to help) in the model. As non-natal subordinates are very rare (only 3.5% of all subordinates are non-natal subordinates; Groenewoud et al. 2018¹⁶; M. Hammers, unpublished data), we did not consider this as a factor in the models. The analysis now includes all territories with one or more subordinates (instead of only those with a single subordinate), the helping status of each subordinate (Y/N), sex of the subordinate, age of the subordinate, age of the dominant, territory quality, the number of subordinates in the territory and the interaction between the dominant's age and the subordinate's sex. We have also investigated whether the effects of dominant age could be explained by selective disappearance effects, by including the dominant's longevity (Van de Pol & Verhulst 2006⁵). These models gave similar results and were not better supported by the data than the models that contained simple age effects. Therefore, we report models with simple age effects (i.e. the simpler, more parsimonious model) in the main text, but we also provide the models that include longevity in the supplementary material (Table S7).

Specifically, we have added the following to the Methods and updated the Results:

(L387-396): "Age of the dominant, age of the subordinate (≤ 1 year old vs. 2+ years old), sex of the subordinate, territory quality and the number of subordinates in the territory were included as predictors. Dominant identity, family group, and year were included as random effects. We included an interaction between sex of the subordinate and the dominant's age to test whether the association between the dominant's age and the subordinate's likelihood of helping differed between male and female subordinates. To check if selective disappearance of poor-quality individuals could explain the age-dependent change in helping status, we added longevity of the dominant to the model (i.e. including only individuals that have died within our study period), following Van de Pol & Verhulst⁵. As we found no evidence for selective disappearance effects (Table S7), we report the results from the simpler models."

4. It could also be usefully clarified how the analysis worked once attention was shifted to the effect of female helper presence versus male helper presence. Specifically, should we interpret the more apparent dominant female age effect on female helper presence as evidence that with increasing dominant female age (1) the one subordinate that those dominant females with one subordinate of either sex have is increasingly likely to be a female subordinate that helps (a relationship that could be a product of age-related changes in sex allocation for example), or that (2) the one subordinate that those dominant females with one subordinate female have is increasingly likely to help?

This was unclear in the previous version. It should be clear now (with the new analysis, see our response immediately above to comment 3) that option 2 is the case here. Any subordinate female that is present is more likely to help with increasing age of the dominant female.

5. On a separate note, given the potential for female helpers to inherit the dominant breeding position on the death of their dominant female (presumably?), while it is suggested that female helpers stand to gain fitness benefits from helping their aging mothers to survive, it would be good to consider the possibility that female helpers could instead suffer a substantial direct fitness cost by helping their mothers to survive if doing so reduces their own chances of inheriting the local dominant position.

Seychelles warblers very rarely inherit the territory (3.7%, Eikenaar et al. 2008¹⁷); therefore, we did not consider the costs of helping in terms of a reduced chance to inherit the dominant position in the group. We now mention that territory inheritance is rare in L269-271; “Territory inheritance in the Seychelles warbler is rare (only 3.7% of subordinates inherit a dominant breeding position, Eikenaar et al. 2008), so it is unlikely that inheritance is the main benefit accrued by subordinates.”

SPECIFIC COMMENTS

6. Introduction

The statement that “studies investigating the relationship between sociality and senescence at the intraspecific level are rare” could be usefully accompanied by acknowledgement of the studies that have looked at the impact of sociality on senescence at the intraspecific level in the context of competition rather than cooperation.

We now acknowledge (L35-36) that some studies have investigated the relationship between intraspecific competition and senescence and cited Beirne et al. (2015¹⁸) and Sharp & Clutton-Brock (2011¹⁹).

7. Results

The rarity of male helpers precluded the analysis of the impact of male helpers on changes in telomere length. As such, it would be useful to state the sample sizes for the other analyses with regard to the presence and absence of male helpers and the presence and absence of female helpers. For example, I was left wondering whether there might be no apparent effect of male helpers on dominant survival principally because the sample size of data points with male helpers present is much smaller than the sample size of data points with female helpers present (see above).

Please see our response to comment 1d.

8. Discussion

Line 162 - “our results indicate that that [sic] female helpers help to delay senescence in dominants” would seem to need some toning down given the potential for alternative explanations to account for the findings (see above).

Agreed. We have replaced this sentence with the following: “our results suggest that female helpers may contribute to delay senescence in dominants and that, at the same time, those dominant females acquire more female helpers as they get older.”

9. Line 206-208 – speculation that helpers might also mitigate reproductive senescence among dominants could be usefully replaced with analyses establishing whether this is the case, given that such data are available.

We fully agree that it would be great to analyse how age-dependent reproductive output is affected by helper presence in a future manuscript, but we feel that the very extensive and complex analysis required to do this properly is outside the scope of the current manuscript, and that this topic warrants a manuscript on its own.

10. Lines 214-215 – the manuscript would benefit throughout from a clearer logical distinction being drawn between co-breeding and helping, as it would seem that female ‘helpers’ may do one or the other in any given breeding attempt, leaving it unclear which behavior might be driving the relationships observed if they were indeed causal. Indeed, if helping is defined as the provision of alloparental care, and co-breeding in this species involves putting offspring in the dominant’s nest, it would be good to clarify whether female co-breeders are actually really helpers at all in that particular breeding attempt.

We have now clarified that we considered all female subordinates that helped with incubation or provisioning as ‘helpers’, irrespective of whether they co-bred or not. This is done because co-breeding subordinates always help with incubation and provisioning (Richardson et al. 2003). Also, co-breeding helpers do not discriminate between offspring, so they help all offspring in the nest (Komdeur et al. 2004).

We have clarified the definition of helpers, which includes both co-breeding with alloparental care and providing alloparental care without co-breeding in L273-279: “Further, older (>1 year old) female subordinates often (*ca* 40% in any year) gain direct fitness benefits through co-breeding (laying an egg in the same nest as the dominant female). Co-breeding subordinates always provide alloparental care and do not discriminate between their own or the dominant female’s offspring (i.e. they help all offspring in the nest). Further, previous studies found no evidence for reproductive conflict caused by co-breeding females (Richardson et al. 2002, Komdeur & Richardson 2007, Bebbington et al. 2017, Kingma et al. 2018). Therefore, we considered all subordinates that helped with incubation or provisioning as ‘helpers’, irrespective of whether they co-bred or not.

11. Methods

Lines 337-338 – “We performed this analysis for male and female dominants separately, as these relationships differ between dominants of the two sexes (see results).” The results don’t appear to show a significant effect of the sex of the dominant on the magnitude of the helper effect on change in telomere length. It would be good to show this, by testing for the 3 way interaction: dominant age x female helper presence x dominant sex. Likewise for the similar statement for the helper presence analysis (lines 350-352).

All analyses are now performed separately for male and female dominants and we do not statistically test for differences in the impact of helpers on male and female dominants.

Reviewer #3 (Remarks to the Author):

This manuscript contains important new information from a classic long-term of Seychelles warblers. Among other results the authors present evidence that: (i) female but not male helpers enhance annual survival of the dominant female; (ii) dominant females but not males suffer telomere shortening if unhelped, but telomere lengthening if helped; and (iii) the likelihood that a subordinate helps (provisions at the nest) increases for female but not male helpers with the age of the dominant female but not the dominant male.

Taken alone, these results are important and likely to attract considerable interest. The paper is mostly clearly written and is very well-illustrated. It is remarkably free of typographical errors, and the data are for the most part appropriately analysed. It is also commendable in being based on a very substantial body of field work (albeit in a nice place).

We are pleased that the reviewer is positive about our manuscript and highlights the importance of these results, and considers that the manuscript is clearly written, well-illustrated and that the data are appropriately analysed.

1. Despite this, and my having read a reasonable proportion (though by no means all) of the Seychelles warbler papers, I found myself repeatedly asking ‘but what if ...’ and does this happen ...’ questions, which make me feel that a very strong manuscript would become even better through some more definitive description of the helping system, and an elaboration and testing of the predictions associated with the models sketched in the second paragraph of p 3.

As I understand the system, the birds occupy territories, on which it is usually possible to identify a dominant pair, but there are also supernumerary birds. While the overwhelming majority of supernumerary males do not help, supernumerary females can (i) live on the territory without helping; (ii) help by feeding the offspring of the dominant female without direct contributions to reproduction, or (iii) co-breed by laying one of the eggs in what becomes a two-egg brood. It seems that the helping birds are often retained but on p. 3 mention is also made of attracting helpers, and the different expectations that then arise if the supernumerary birds are kin. I think the manuscript would be vastly improved if we knew the age structure, kinship to the dominant, and proportion and type of subordinates that have dispersed or remained philopatric, as these can provide further evidence of the authors’ conceptual model.

Thank you for this comment. We agree and we have provided a much more detailed description of the helping system in the Methods, which includes details on age-structure, kinship, inheritance and the frequency of subordinates dispersing to a new subordinate position. All this information is published in other papers and this manuscript would become too long if this was all included here (it is not a full review of the warbler study system, for this see Komdeur et al. 2016²⁰), so we included a detailed summary of the information in L250-287.

2. For example, in Fig 4 there is evidence that old females but not old males are increasingly likely to have helpers (it is not clear - and should be - how co-breeders are treated here). This is consistent with the authors’ conceptual model. However, if it is due to helpers benefiting from assisting kin to resist senescence it poses some expectations on the knowledge that the helpers have about the social milieu in which they live. Most importantly, there has to be a recognition by the potential helper that the mother is old. There would possibly be direct knowledge that this is true if the potential helper had been on the territory herself for many years, and would be supported by an increase in help by old helpers, but the relevant data are not given. However, relatedness to the male being assisted is likely to decline with age because the female changes her social or extra-pair partner. The second possibility is an increase in ‘staying’ incentives by old dominants. This presumably predicts an increase in co-breeding with the age of the mother, but the relevant data are not given, though again they are presumably available, and I suspect in this case there may be alternative explanations.

Both subordinates that co-breed (and always help), and subordinates that do not co-breed but do help are termed helpers because they both provide alloparental care and because reproductive conflict between dominant females and co-breeding subordinates is generally absent in this system (Komdeur &

Richardson 2007). We have clarified this in the methods section (L277-278); please also see our response to comment 1c of reviewer 2.

Since female subordinates often stay in a territory for multiple years and female subordinates are often older than male subordinates (see Figure below), it appears likely that the female subordinates can assess the age of the mother. Our improved analysis of helping propensity (L378-392) shows that older subordinates are more likely to help (Table 3). Furthermore, a new analysis we performed to investigate the likelihood that a subordinate female reproduced (L402-411) showed that older female subordinates are much more likely to co-breed than younger female subordinates, but also that the likelihood of reproduction by the subordinate is not significantly associated with the age of the mother (L133-136; Table S4).

Frequencies of male and female helpers of different ages in our dataset.

3. The other issue is the untested notion that a female queuing for dominance gains a lifetime fitness benefit from prolonging the life of the bird at the top of the queue, and hence delaying their eventual succession. More information on the pattern of succession would be helpful, but I suspect some modelling would be necessary here.

We agree that modelling the inclusive fitness benefits of helping older dominants would be interesting, but this is beyond the scope of this manuscript. However, we do now provide more information on the benefits that subordinates gain from staying in the territory and on the consequences for dispersal if parents are replaced by unrelated stepparents. Eikenaar et al. (2007) provided evidence that the presence of parents in the natal territory may promote delayed dispersal and facilitate the eventual acquisition of a breeder position outside the natal territory in the Seychelles warbler. However, actual territory inheritance in the Seychelles warbler is rare: only 3.7% of subordinates inherit a dominant breeding position (Eikenaar et al. 2008). Finally, there is evidence that unrelated stepparents evict unrelated subordinates from their territory (Eikenaar et al. 2007; Kingma et al 2016). We have added this, and

more detailed information on the benefits obtained by subordinates, to the Methods (L269-287).

Minor comments

4. Why are model predictions illustrated in this Figure, but raw means in the other Figures

This was our error. All figures now contain the raw means.

5. It is uncertain what is meant by the sample sizes in Figure 4. A single sample size implies that each female has both a male and female subordinate, which seems unlikely to be true.

We now provide sample sizes for male and female subordinates separately.

REFERENCES

- 1 Komdeur, J., Richardson, D. S. & Burke, T. Experimental evidence that kin discrimination in the Seychelles warbler is based on association and not on genetic relatedness. *Proc. R. Soc. B* **271**, 963-969 (2004).
- 2 Brouwer, L., Richardson, D. S., Eikenaar, C. & Komdeur, J. The role of group size and environmental factors on survival in a cooperatively breeding tropical passerine. *J. Anim. Ecol.* **75**, 1321-1329 (2006).
- 3 Hammers, M., Richardson, D., Burke, T. & Komdeur, J. The impact of reproductive investment and early-life environmental conditions on senescence: support for the disposable soma hypothesis. *J. Evol. Biol.* **26**, 1999-2007 (2013).
- 4 Wood, S. & Scheipl, F. gamm4: Generalized additive mixed models using mgcv and lme4. (2015).
- 5 van de Pol, M. & Verhulst, S. Age-dependent traits: a new statistical model to separate within- and between-individual effects. *Am. Nat.* **167**, 766-773 (2006).
- 6 van Boheemen, L. A. *et al.* Compensatory and additive helper effects in the cooperatively breeding Seychelles warbler (*Acrocephalus sechellensis*). *Ecology & Evolution* (in press).
- 7 Cockburn, A. *et al.* Can we measure the benefits of help in cooperatively breeding birds: the case of superb fairy-wrens *Malurus cyaneus*? *J. Anim. Ecol.* **77**, 430-438 (2008).
- 8 Eikenaar, C., Richardson, D. S., Brouwer, L. & Komdeur, J. Parent presence, delayed dispersal, and territory acquisition in the Seychelles warbler. *Behav. Ecol.* **18**, 874-879 (2007).
- 9 Eikenaar, C., Brouwer, L., Komdeur, J. & Richardson, D. S. Sex biased natal dispersal is not a fixed trait in a stable population of Seychelles warblers. *Behaviour* **147**, 1577-1590 (2010).
- 10 Komdeur, J. & Pels, M. D. Rescue of the Seychelles warbler on Cousin Island, Seychelles: the role of habitat restoration. *Biol. Conserv.* **124**, 15-26 (2005).
- 11 Richardson, D. S., Burke, T. & Komdeur, J. Direct benefits and the evolution of female-biased cooperative breeding in Seychelles warblers. *Evolution* **56**, 2313-2321 (2002).
- 12 Komdeur, J. & Richardson, D. S. Molecular ecology reveals the hidden complexities of the Seychelles warbler. *Adv. Study Behav.* **37**, 147-187 (2007).
- 13 Kingma, S. A. *et al.* in *PhD thesis F. Groenewoud* (ed Frank Groenewoud) (University of Groningen, 2018).
- 14 Komdeur, J. Experimental evidence for helping and hindering by previous offspring in the cooperative-breeding Seychelles warbler *Acrocephalus sechellensis*. *Behav. Ecol. Sociobiol.* **34**, 175-186 (1994).
- 15 Bebbington, K. *et al.* Joint care can outweigh costs of nonkin competition in communal breeders. *Behav. Ecol.* **29**, 169-178 (2017).
- 16 Groenewoud, F. *et al.* Subordinate females in the cooperatively breeding Seychelles warbler obtain direct benefits by joining unrelated groups. *J. Anim. Ecol.* (2018).
- 17 Eikenaar, C., Komdeur, J. & Richardson, D. S. Natal dispersal patterns are not associated with inbreeding avoidance in the Seychelles warbler. *J. Evol. Biol.* **21**, 1106-1116 (2008).
- 18 Beirne, C., Delahay, R. & Young, A. Sex differences in senescence: the role of intra-sexual competition in early adulthood. *Proc. R. Soc. B* **282**, 20151086 (2015).
- 19 Sharp, S. P. & Clutton-Brock, T. H. Reluctant challengers: why do subordinate female meerkats rarely displace their dominant mothers? *Behav. Ecol.* **22**, 1337-1343 (2011).
- 20 Komdeur, J., Burke, T., Dugdale, H. L. & Richardson, D. S. Seychelles warblers: Complexities of the helping paradox. *Cooperative breeding in vertebrates. Studies of ecology, evolution and behaviour.* Cambridge Univ. Press, Cambridge, UK, 197-216 (2016).

Reviewers' Comments:

Reviewer #1:

Remarks to the Author:

The authors have addressed all my comments and I have no further comments. I think this is a very nice contribution.

Reviewer #2:

Remarks to the Author:

The authors have responded thoroughly to my comments, strengthening the manuscript in the process. A few points could now be usefully addressed to further strength the manuscript.

1. Evidence of load-lightening

The new analysis suggesting that dominant females invest less in incubation when assisted by a female helper is welcome, and strengthens the manuscript (along with the findings of the parallel paper on provisioning now in press). The incubation analysis presented is currently underdeveloped though, with no covariate predictors being fitted alongside helper presence to control for alternative potentially confounding sources of variation in dominant female incubation effort (e.g. group size effects, dominant female age, territory quality etc), and the data is available to fit these. Please extend this analysis with a view to demonstrating that this finding is robust to controlling for such factors.

2. 'Helpers' are a mix of co-breeders and non-breeders, which is important.

It was good to see this issue clarified in the manuscript. As this terminology has the potential to cause confusion in the downstream interpretation of this paper by the field, two changes could usefully be made to minimise the chance of this happening:

(i) As the authors do not currently tease apart whether the effects arise from effects of co-breeding or alloparental helping, it would be good to explicitly state that this is the case in the discussion of the paper.

(ii) It would seem essential to acknowledge in the abstract that 'helpers' actually include both co-breeders and alloparental non-breeding helpers, otherwise many will naturally interpret this as evidence that non-breeding helpers delay senescence, when the patterns might actually be driven by co-breeders (which might commonly be termed 'breeders' rather than 'helpers').

3. The title of the manuscript is currently "Helpers delay parental senescence: evidence from a cooperatively breeding bird". I would suggest rephrasing it for two reasons...

(i) The first part is a causal statement, yet no evidence of a causal link between the two has been presented in the manuscript. The manuscript highlights via a range of (now quite compelling) correlative evidence that 'helpers' may indeed delay 'parental' senescence, but we can't conclude from this that "helpers delay parental senescence".

(ii) As per point 2 above, the use of 'helpers' and 'parents' is problematic here, because the distinction between these two roles is not clear in this paper (helpers are frequently parents too), leaving the statement rather non-sensical.

4. The rationale underpinning the author's hypothesised link between helping and senescence might be usefully clarified in places. For example, the opening line of the abstract "Cooperation among group members is predicted to favour delayed senescence..." implies that the authors are investigating here how helping will impact patterns of selection on senescence (e.g. that helping might select for increased investment in self-maintenance among dominants, thereby delaying senescence), but the

rationale being used to explain the findings is that, within a generation, helpers may allow dominants to invest more in self-maintenance / maintaining condition etc because they lighten the workloads of dominants (in which there is no need to invoke effects of helping on patterns of selection on self-maintenance / senescence, though such effects may of course also exist). Similarly, the logic seems focussed more on patterns of selection in the changes made to the manuscript in response to Reviewer 1's point 9, lines 236-242 e.g. "... means the selection on delayed senescence may..." etc. Both arguments can of course be made, but it might just help to clarify things if there was greater logical consistency between the rationale outlined in the setup of the study (at least in the abstract, but perhaps elsewhere too) and the core rationale used to interpret the findings. Just a suggestion.

Reviewer #3:

Remarks to the Author:

This manuscript is improved by the revisions made by the authors, which have been carefully made in response to the criticisms and questions levelled by the three referees. In particular, some speculative assertions and overinterpretation of complex models has been reduced.

The manuscript does suffer from the modern tendency of stacking the SI section with the responses to the comments of the referees. While this may be inevitable in journals like the Nature stable and its ilk, I think that the manuscript for maximum impact should contain the key Figures in the main text. In particular, I would recommend including Supplementary Figure 1 and 2 in the main text.

I think lines 150 to 156 overstate the generality of these patterns among cooperatively breeding birds. For this body size, the survival of the Seychelles warbler is itself exceptional.

In relation to my original comments, I (Referee 3) am still sceptical of the evidence provided that females know the age of their mother, as while the tenancies of the long-lived females are remarkable, what you actually need to know is the age distribution of the helper set aiding each age class of the mother.

Secondly, I found it very interesting that only 4% of helpers become dominant, but the other statistic I would have liked is the proportion of dominants still on their natal territory.

We are grateful that all three reviewers are pleased with the revision.

REVIEWERS' COMMENTS:

Reviewer #1 (Remarks to the Author):

The authors have addressed all my comments and I have no further comments. I think this is a very nice contribution.

We are pleased that the reviewer finds that we have dealt satisfactorily with all comments.

Reviewer #2 (Remarks to the Author):

The authors have responded thoroughly to my comments, strengthening the manuscript in the process. A few points could now be usefully addressed to further strength the manuscript.

We are pleased that the reviewer finds that we have responded thoroughly to the comments and that the manuscript was strengthened by the revisions. We also appreciate the additional comments provided here, as we feel that the changes based on these have further improved our manuscript.

1. Evidence of load-lightening

The new analysis suggesting that dominant females invest less in incubation when assisted by a female helper is welcome, and strengthens the manuscript (along with the findings of the parallel paper on provisioning now in press). The incubation analysis presented is currently underdeveloped though, with no covariate predictors being fitted alongside helper presence to control for alternative potentially confounding sources of variation in dominant female incubation effort (e.g. group size effects, dominant female age, territory quality etc), and the data is available to fit these. Please extend this analysis with a view to demonstrating that this finding is robust to controlling for such factors.

We have further developed the incubation analysis and included group size, dominant female age, female age², and territory quality (i.e. the same predictors as in the other analyses). The results of this analysis are similar and show that these results are robust to the inclusion of these factors. We now include the methods and results of this analysis in the main text instead of in the supplementary materials.

2. 'Helpers' are a mix of co-breeders and non-breeders, which is important.

It was good to see this issue clarified in the manuscript. As this terminology has the potential to cause confusion in the downstream interpretation of this paper by the field, two changes could usefully be made to minimise the chance of this happening:

(i) As the authors do not currently tease apart whether the effects arise from effects of co-breeding or alloparental helping, it would be good to explicitly state that this is the case in the discussion of the paper.

(ii) It would seem essential to acknowledge in the abstract that ‘helpers’ actually include both co-breeders and alloparental non-breeding helpers, otherwise many will naturally interpret this as evidence that non-breeding helpers delay senescence, when the patterns might actually be driven by co-breeders (which might commonly be termed ‘breeders’ rather than ‘helpers’).

Agreed.

(i) Following the suggestion of the reviewer we have added the following sentence to the discussion (L232-235): “Here we do not tease apart whether the effects outlined above arise from co-breeding or alloparental helping as separating these two types of helpers is difficult in this system, given that some non-breeders may be individuals that have attempted to breed, but failed to do so successfully.”

(ii) We have added “(both co-breeders and non-breeding helpers)” to the abstract (L25).

3. The title of the manuscript is currently “Helpers delay parental senescence: evidence from a cooperatively breeding bird”. I would suggest rephrasing it for two reasons...

(i) The first part is a causal statement, yet no evidence of a causal link between the two has been presented in the manuscript. The manuscript highlights via a range of (now quite compelling) correlative evidence that ‘helpers’ may indeed delay ‘parental’ senescence, but we can’t conclude from this that “helpers delay parental senescence”.

(ii) As per point 2 above, the use of ‘helpers’ and ‘parents’ is problematic here, because the distinction between these two roles is not clear in this paper (helpers are frequently parents too), leaving the statement rather non-sensical.

Agreed. We have revised the original title so that it does not include (i) a causal statement (ii) or included the word ‘parental’. The new title of the manuscript is: “Breeders that receive help age more slowly in a cooperatively breeding bird”

4. The rationale underpinning the author’s hypothesised link between helping and senescence might be usefully clarified in places. For example, the opening line of the abstract “Cooperation among group members is predicted to favour delayed senescence...” implies that the authors are investigating here how helping will impact patterns of selection on senescence (e.g. that helping might select for increased investment in self-maintenance among dominants, thereby delaying senescence), but the rationale being used to explain the findings is that, within a generation, helpers may allow dominants to invest more in self-maintenance / maintaining condition etc because they lighten the workloads of dominants (in which there is no need to invoke effects of helping on patterns of selection on self-maintenance / senescence, though such effects may of course also exist). Similarly, the logic seems focussed more on patterns of selection in the changes made to the manuscript in response to Reviewer 1’s point 9, lines 236-242 e.g. “... means the selection on delayed senescence may...” etc. Both arguments can of course be made, but it might just help to clarify things if there was greater logical consistency between the rationale outlined in the setup of the study (at least in the abstract, but perhaps elsewhere too) and the core rationale used to interpret the findings. Just a suggestion.

We appreciate this suggestion and have carefully gone through the manuscript to clarify the rationale underpinning the link between helping and senescence in breeders. We have made the following changes:

- We followed the suggestion of the reviewer and replaced “favour” with “lead to” in the abstract to avoid implying that helpers affect patterns of selection.
- In L55 with replaced “strong selection on” with “a strong incentive for”
- In L236-242 (L268-273 in the current version of the manuscript), we discuss the broader implications of our study and discuss the potential implications of a within-generation effect as found in our study for selection on delayed senescence and cooperation. We have added the following sentence to the start of this paragraph: “In the longer term, helping-enabled improvements in the late-life survival of dominants may drive the evolution of longer lifespan in cooperative breeders, but this prediction remains to be tested.”

Reviewer #3 (Remarks to the Author):

This manuscript is improved by the revisions made by the authors, which have been carefully made in response to the criticisms and questions levelled by the three referees. In particular, some speculative assertions and overinterpretation of complex models has been reduced.

We are pleased that the reviewer finds that the manuscript is improved and are grateful for the additional comments, to which we respond below.

The manuscript does suffer from the modern tendency of stacking the SI section with the responses to the comments of the referees. While this may be inevitable in journals like the Nature stable and its ilk, I think that the manuscript for maximum impact should contain the key Figures in the main text. In particular, I would recommend including Supplementary Figure 1 and 2 in the main text.

We have now included Supplementary Figure 2 and Supplementary Table 3 to the main text (Figure 4 and Table 4) of the manuscript so that the manuscript contains all key figures and tables. We have not included Supplementary Figure 1 (which has now become Supplementary Figure 2) because most information in this figure is already presented in Figure 4 in the main text.

I think lines 150 to 156 overstate the generality of these patterns among cooperatively breeding birds. For thie (sic) body size, the survival of the Seychelles warbler is itself exceptional.

We agree and have added “in the Seychelles warbler” to clarify that this statement is specifically about the Seychelles warbler, not about cooperatively breeding birds in general.

In relation to my original comments, I (Referee 3) am still sceptical of the evidence provided that females know the age of their mother, as while the tenancies of the long-lived females are remarkable, what you actually need to know ids the age distribution of the helper set aiding each age class of the mother.

Given the reviewer’s comment we now state in the discussion that we do not know the actual mechanism behind why subordinates are more likely to stay and help when female dominants

are older (L244). Indeed, for the results and conclusions of this study it is not directly important whether subordinates are able to assess the age of the dominants or not, nor do we claim that subordinates are able to assess the ages of the dominants. For dominants, the decision to retain or evict a subordinate depends on the fitness benefits of retaining the subordinate versus the costs of not doing so, which is independent of whether a subordinate can assess the dominant's age or not. As indicated in the introduction and discussion, we expect that as the dominant's fitness declines because of senescence, they may become more likely to try to retain (or at least not evict) subordinates and provide incentives (e.g. a share in reproduction) to encourage the subordinate to stay and help. For a subordinate, the decision to stay and help is dependent on the benefits that they may obtain in the resident territory (e.g. indirect and direct fitness benefits) versus the benefits of leaving the territory. It is unknown whether it is important for them to be able assess the age of the dominants.

Secondly, I found it very interesting that only 4% of helpers become dominant, but the other statistic I would have liked is the proportion of dominants still on their natal territory.

This statistic was already in our manuscript (L269-271). We have rephrased it to make it clearer and the sentence now reads: "Territory inheritance in the Seychelles warbler is rare (only 3.7% of dominant breeding positions are obtained via offspring inheriting this status on their natal territory⁵⁸)".